# Geometrical congruence, greedy navigability and myopic transfer in complex networks and brain connectomes

Carlo Vittorio Cannistraci [1,2,3,4,5,6] & Alessandro Muscoloni [1,4]

We introduce in network geometry a measure of *geometrical congruence (GC)* to evaluate the extent a network topology follows an underlying geometry. This requires finding all topological shortest-paths for each nonadjacent node pair in the network: a nontrivial computational task. Hence, we propose an optimized algorithm that reduces 26 years of worst scenario computation to one week parallel computing. Analysing artificial networks with patent geometry we discover that, different from current belief, hyperbolic networks do not show in general high GC and efficient greedy navigability (GN) with respect to the geodesics. The myopic transfer which rules GN works best only when degree-distribution power-law exponent is strictly close to two. Analysing real networks—whose geometry is often latent—GC overcomes GN as marker to differentiate phenotypical states in macroscale structural-MRI brain connectomes, suggesting connectomes might have a latent neurobiological geometry accounting for more information than the visible tridimensional Euclidean.

"There is enough treachery, hatred violence absurdity in the average human being to supply any given army on any given day [...]". Although feelings such as treachery and hatred do not comply with the scope of this study, in the incipit of the poem "The Genious of the crowd", Charles Bukowski offers an impressive lecture on how to leverage the average for analyzing collective behaviors in a complex disordered system such as our society, and the risks to deviate from this average by following an incongruent path of creative and indipendent thinking, which might require significantly larger time. Likewise, in the footsteps of Bukowski's lecture from which we took inspiration, the essence of this study can be summarized in an aphorism that at the moment might result cryptic, but that at the end of this scientific essay should reveal its message: "beware the average geometrical projection of topological shortest paths between two points in a network and the time to compute them". Indeed, to fulfill this aphorism we propose an optimized algorithm that reduces 26

years worst scenario of computation to 1 week. And, finally, we are able to measure the extent to which a network topology is congruent with an associated manifold geometry.

If the shortest paths provide the syntax of the interactions between two points, their projections in the respective underlying geometrical space reveal the semantics: the meaning behind those interactions. Not suprisingly, studies in network geometry[1] suggest that the latent geometrical space behind the observable topology of a network drives the navigation[2] of the network structure according to the distances between nodes in the geometrical space[3]. This means that the process behind the formation of connectivity in many networked complex systems follows a rule of association between the latent variables of the system that can be described by a manifold in a geometrical space. This geometrical space determines also the type of network. In relation to the global curvature of the manifold we can have three different types of geometries: hyperbolic (negative

[1]Center for Complex Network Intelligence (CCNI), Tsinghua Laboratory of Brain and Intelligence (THBI), Tsinghua University, Beijing, China. [2]Department of Computer Science, Tsinghua University, Beijing, China. [3]Department of Biomedical Engineering, Tsinghua University, Beijing, China. [4]Biomedical Cybernetics Group, Biotechnology Center (BIOTEC), Center for Molecular and Cellular Bioengineering (CMCB), Technische Universität Dresden, Dresden, Germany. [5]Department of Physics, Technische Universität Dresden, Dresden, Germany. [6]Center for Systems Biology Dresden (CSBD), Dresden, Germany. ✉e-mail: kalokagathos.agon@gmail.com

curvature), Euclidean (zero curvature), elliptic (positive curvature). Likewise, in network geometry we can encounter: hyperbolic networks, Euclidean networks and elliptic networks. Different geometries can trigger different structural and functional features of the associated networks.

Network geometry is currently rising as a compelling research area in physics[1]. The question of how network geometry influences network navigation is a crucial topic in science and engineering, and a recent review[1] (that might soon become an essential reference for the field of network geometry) reports a list of theoretical studies according to which *hyperbolic networks* are maximally efficient for geometric navigation[4,5]. The main reason behind this phenomenon is assumed the proximity of topological shortest paths (TSP) in the hyperbolic networks to the corresponding geodesics in the underlying hyperbolic geometry. If we enucleate better this concept, this means that the projections of the topological shortest paths (pTSP) follow closely their associated hyperbolic geodesics in the underlying space[1,4] and, since in network science this is defined as geometrical congruence of the network topology with the underlying geometry, hyperbolic networks are believed geometrically congruent[1]. Another key property for efficient geometrical navigation is the existence of super-hubs that interconnect large parts of the network. This happens when the network degree distribution follows a power-law with exponent $\gamma < 3$[1,4], in which case scale-free[6] networks are termed ultra-small-world[1,7].

Greedy navigation[2] is the result of a type of local phenomenon that we define myopic transfer from a source node to a destination node. The fact that the transfer is myopic (near-sighted) means that a node can see only the geometrical distances of its neighbors to the destination and can proceed transferring to that neighbor node which is closest to the destination, creating what is called a greedy routing path (GRP). Boguñá et al.[8] proposed a theoretical demonstration that greedy navigation in networks with $\gamma < 3$ and strong clustering (such as hyperbolic networks[4]) can always find these ultrashort paths which follow the geodesics[1,8], and thus navigation in hyperbolic networks with $\gamma < 3$ is believed maximally efficient because of their supposed geometrical congruence[1,4]. Vice versa, a network that is maximally navigable by design is considered similar to hyperbolic networks[1].

Despite the abovementioned theoretical research, we have not found any study with computational evidences that validate these theoretical conclusions by means of numerical simulations that measure the level of congruence on hyperbolic network models. This might be due to two factors: (i) the lack of definition of a computational measure of geometrical congruence in network science; (ii) the difficulty to design an efficient (reasonable time complexity) algorithm to measure all the shortest paths (and their projections) between each nonadjacent node pair in the network. Here, we address these problems: (i) proposing a measure of geometrical congruence of a network topology to a reference geometry, which can be expressed as geometrical distances or network weights (generated by a latent geometry); (ii) proposing an innovative algorithm that drastically reduces the time complexity. Then, we proceed to measure geometrical congruence and greedy navigability efficiency in networks with patent (the contrary of latent in *Latin*) geometrical space, in the sense that the coordinates of the network nodes in this space are known; as opposed to a latent geometrical space in which the coordinates of the network nodes are unknown. Specifically, we investigate hyperbolic networks generated with the nonuniform popularity-similarity optimization (nPSO) model[9,10] which is a generalization of the PSO model[11] able to grow realistic hyperbolic soft random geometric graphs with tailored community structure. Finally, we show how, contrary to the greedy navigability measures that need nodes geometrical coordinates (patent geometrical space, known or inferred by network embedding), the proposed geometrical congruence measure can be applied as a marker to reveal differences in real networks with latent geometrical

space such as brain connectomes, without the requirement of nodes geometrical coordinates, but only geometrical weights (presumably associated with the latent space). Indeed, brain connectomes topological weights are associated with a developmental neurobiological geometry which, according to some studies, might be approximated by hyperbolic distances[12–14]. However, as a matter of fact, the geometry is latent because both the type of geometry and the coordinates of the nodes in this developmental neurobiological space are unknown.

## Results

### Types of paths and associated lengths in the topological and geometrical domains

This section aims to introduce the conceptual and visual differences between the types of paths, lengths and distances we adopt in this study. For instance, and without loss of generality, let us consider a network that is generated according to an underlying geometry that is hyperbolic: a hyperbolic network. We generate it by using the nPSO hyperbolic network model, which has the following parameters: nodes size N; average degree $\bar{d}$; power-law exponent γ (generally between 2 and 3, under value 3 the network is considered to acquire a marked heterogeneous and hierarchical structure); temperature T (generally between 0 and 1, the closer to 0 the more the network is clustered, the closer to 1 the more is random); number of tailored community C (a value of 0 means no tailored community which corresponds with the standard PSO, value of 1 is not used, any value larger than 1 indicates the number of tailored communities). More specifically, in Fig. 1 we display a nPSO hyperbolic network (with $N = 100$, $\bar{d} = 8$, $\gamma = 2.5$, $T = 0.1$ and $C = 5$) in the native disk representation of the hyperbolic space (see Code Availability section for the code to reproduce such representations). Figure 1a displays the network in the hyperbolic disk where the links are drawn in gray color according to the hyperbolic geodesics.

The greedy routing path (GRP, Fig. 1b, c yellow path) is a directed path that is generated by a local phenomenon we term myopic transfer from a source node to a destination node in the network. The fact that the transfer is myopic (near-sighted) means that a node can see only the geometrical distances of its neighbors to the destination and can proceed transferring to that neighbor node which is closest to the destination. Hence, in order to compute the GRP, geometrical knowledge is necessary. The GRP is greedy because it is the result of a local process that can approximate but does not guarantee to find the shortest path between two nodes. This means that, although a network is connected in one unique component (as the ones in our study), there is not guarantee that a GRP exists (in the sense that it terminates arriving to destination) for each pair of nodes: some GRPs can get stuck in local minima[4] and do not successfully reach the destination node. Figure 1b offers an example of a successful GRP (yellow path), whereas Fig. 1c of an unsuccessful one (yellow path). This happens because the myopic transfer (also known as greedy forwarding[4]) is often inefficient[4] and the GRPs might fail (unsuccessful GRP) to arrive to destination when a loop is formed because a packet returns to the previously visited node along the path. The GRP is associated with the greedy routing navigability of a network and more formal details are provided in the next section. When a GRP is successful to arrive to destination, it is unique, and the geometrical length of its projection (pGRP) in the underlying geometrical space—sum of the distances of the connections involved in such path—is used as a measure that approximates the distance between two nodes in the graph. To note that the GRP is a directed path that is asymmetric, meaning that going from a node $i$ to a node $j$ via a GRP can be different that going from $j$ to $i$.

The topological shortest path (TSP, Fig. 1b dashed black lines) is the path formed by the smallest number of hops that connects two nodes in the unweighted network, this means that its computation does not require knowledge of the underlying geometry. The TSP can be shorter or equal to the GRP, but at least one or multiple TSPs always

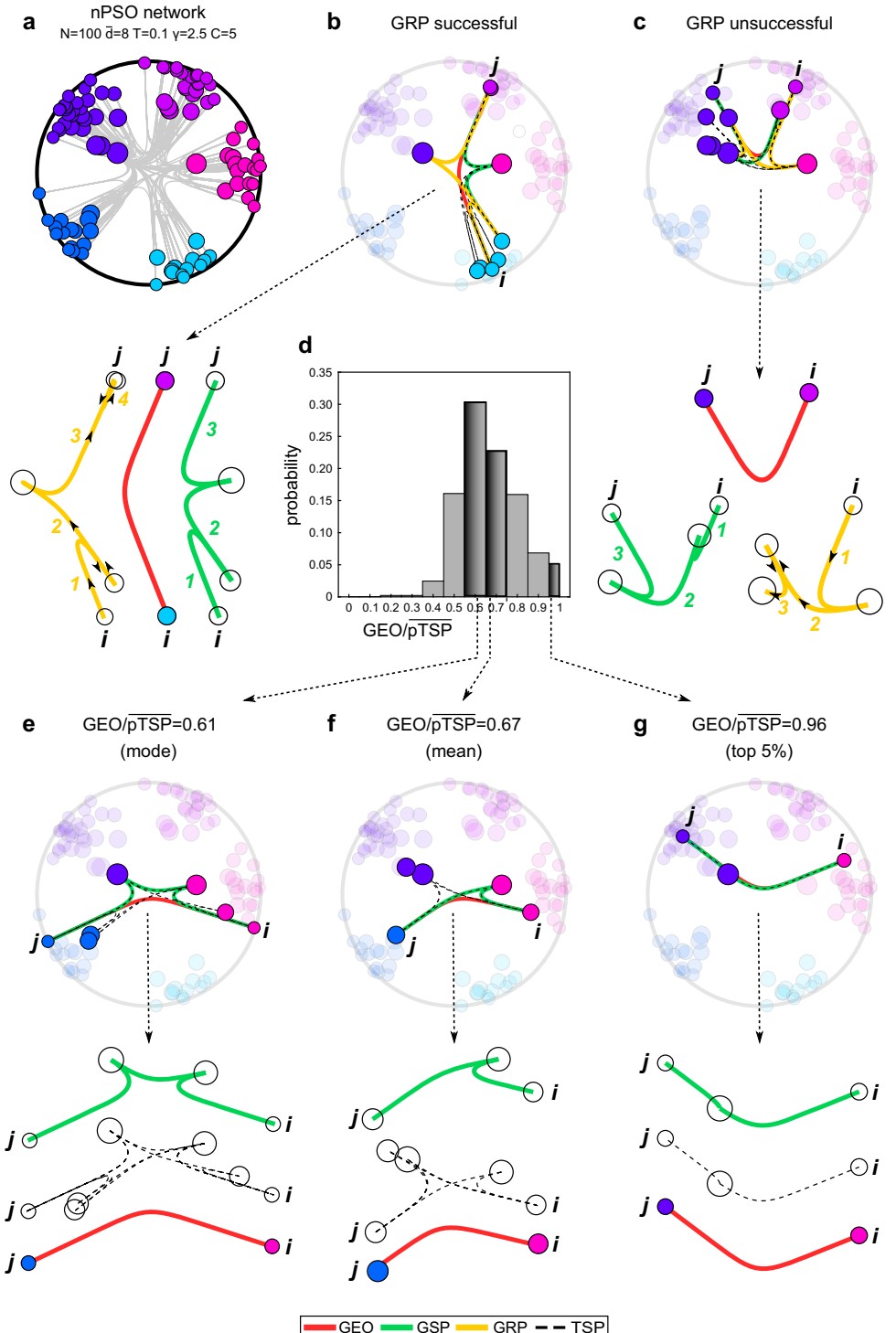

**Fig. 1 | Visualization of the different types of paths in the geometrical (hyperbolic) space.** A nPSO network has been generated with parameters $N = 100$, $\bar{d} = 8$, $T = 0.1$, $\gamma = 2.5$ and $C = 5$. **a** Representation of the nPSO network in the native disk representation of the hyperbolic space: the links are in grey colour and follow the hyperbolic geodesics, the nodes are colored by community membership and their size is proportional to the logarithm of the degree. **b** The panel highlights the hyperbolic geodesic (GEO, red), the geometrical shortest path (GSP, green), the greedy routing path (GRP, yellow with arrows of directionality) and the hyperbolic links involved in all the possible topological shortest paths (TSP, black dashed) between two specific nonadjacent nodes in the network ($i$ and $j$), characterized by a successful GRP. The exploded-view representation (on the left) of

GEO, GSP and GRP emphasizes the diversity of these paths, the number of path's hops and the directionality of GRP is indicated. **c** Analogous to panel (**b**) but for two nonadjacent nodes characterized by an unsuccessful GRP. **d** The histogram indicates, at different intervals, the proportion of nonadjacent node pairs within a certain range of GEO/$\overline{pTSP}$ values in the nPSO network. **e** The panel highlights GEO, GSP, and TSP between two nonadjacent nodes characterized by a ratio GEO/$\overline{pTSP}$ = 0.61 corresponding to the mode value. **f** Analogous to panel (**e**) but for two nonadjacent nodes characterized by a ratio GEO/$\overline{pTSP}$ = 0.67 corresponding to the mean value. **g** Analogous to panel (**e**) but for two nonadjacent nodes characterized by a ratio GEO/$\overline{pTSP}$ = 0.96 corresponding to a value in the top 5% of the histogram.

exist between a given pair of nodes, differently from the GRP that if exists is unique. The hop length of the TSP can be computed for instance using Dijkstra's algorithm[15]. However, detecting and enumerating all the possible TSPs associated with each pair of nodes in the networks is a not trivial computational task and, in this study, we propose an efficient algorithm to address this problem. Because of this, we can progress to compute the exact congruence of a network topology with an underlying geometrical space by measuring the average deviation of the pTSPs from the associated geodesic. The pTSP is the geometrical length of the projection of a TSP in the underlying geometrical space and approximates the distance between two nodes in the graph.

The geometrical shortest path (GSP, Fig. 1b, green line) is the weighted shortest path computed using for instance the Johnson's algorithm[16] on the weighted network. The connections can be weighted by the geometrical distances between the node pairs, therefore the length of the GSP is the sum of the geometrical distances of the hops involved in such path. Note that the pTSP is also a sum of the geometrical distances of the hops involved in the path, but in this case the path is the TSP, which is computed on the unweighted network. We clarify that if the GSP and the TSP have the same number of hops, then the length of the GSP is equivalent to the smallest pTSP. However, the GSP might also have a larger number of hops than the TSP and, in that case, there would not be equivalence. For instance, there might be a GSP as sum of the weights associated with 3 hops, that in total is shorter than the pTSP associated with a TSP of 2 hops. Differently from the GRP that is unique, but it might not exist for some node pairs, there exists always at least one GSP between a node pair.

The geodesic (GEO, Fig. 1b, red line) is the geometrical distance between a pair of network nodes in the underlying geometrical space. Finally, we stress that the GRP is the only path in this section that: (i) is directed, (ii) can be asymmetric and (iii) can unsuccessfully stop before arriving to destination.

## From myopic transfer and greedy navigability to the geometrical congruence

In this section we will introduce the greedy routing navigability (sometime simplified as greedy routing or greedy navigability) and the measures currently available to evaluate its performance. Let us consider a scenario in which any node of a network is not aware of the global network structure but can only see its first neighbor nodes, therefore having a very restricted local topological knowledge. How can such node transfer packets (of information, matter, energy, etc.) to specific destination nodes efficiently without global topological knowledge of the network structure? In brief, the greedy routing navigability problem deals with understanding and evaluating how specific topological arrangements of the network can efficiently guide myopic transfer (in the sense that is near-sighted) from a source node to a destination node. We propose the definition of myopic transfer in the previous sections of this study, in order to name the type of local phenomenon that gives rise to the greedy routing path (GRP).

Following the seminal work of Kleinberg[2], that in 2000 introduced the concept of greedy routing (GR) navigability in small-word networks, in 2008 Boguñá, Krioukov and colleagues[3] discovered that the geometry underlying the network topology is a fundamental property that modifies greedy routing navigability. The underlying geometry, which describes the rule of association between the latent variables in a complex connected system, can shape the network structural connectivity in a way that can facilitate transfer at distance without seeing at distance, giving rise to this local phenomenon that we suggest to name myopic transfer. Note that Krioukov et al.[4] used in their study the term greedy forwarding to define this local transfer. We prefer to use the expression myopic transfer for two reasons. First, the word greedy refers to the type of algorithm but does not conceptually convey the near-sighted topological knowledge in which a node lies when it needs

to transfer the packet. Second, the word forwarding might induce to think that the packet is forwarded towards destination, whereas in reality this type of local transfer is often inefficient[4] and does not guarantee that in general the packet is forwarded to destination, but it can also return back remaining stuck. Peculiar topological shapes of a network carved by nonlinear geometries can facilitate myopic transfer across some specific directions of the space (topological corridors) while creating barriers in other directions[3].

Krioukov et al.[4] proposed several measures to evaluate greedy routing navigability in hyperbolic networks, whose characteristics and limitations are discussed in detail in Supplementary Note 1: "Previous measures for evaluation of greedy routing navigability and their limitations." In Supplementary Note 1, we also provide technical details on how to define and test measures for greedy routing navigability. Here for brevity we discuss the most relevant of these measures that are all based on the GRPs.

The first is the success ratio, which expresses the proportion (or percentage) of successful GRPs that reach their destinations. The second is the hyperbolic stretch, named "stretch" because always positive. In particular, there are two types of hyperbolic stretches (S2 and S3), which express respectively the average hyperbolic deviations of the successful GRPs (S2) or the associated GSPs (S3) with respect to the GEO. Associated GSPs mean that only GSPs between node pairs for which a successful GRP exists are considered. Indeed, these hyperbolic stretches account exclusively for a part of the network topology because they are defined only for the nonadjacent nodes pairs that are connected by a successful GRP. Therefore, they are well-defined to evaluate greedy navigability only when the success GRP ratio is high. This implies that they are not suitable to evaluate greedy navigability in general in complex networks, because according Krioukov et al.[4] (text is reported verbatim): ≪These GF processes [note: GF stays for greedy forwarding, which in this study we refer as myopic transfer] can be very inefficient. They can often get stuck at local minima, or even if they succeed reaching the destination, they can travel along paths much longer than the optimal shortest paths available in the network.≫

Nevertheless, Krioukov et al.[4] in same study claims (text is reported verbatim): ≪The lower these two stretches [S2 and S3], the closer the greedy and shortest paths stay to the hyperbolic geodesics, and the more congruent we say the network topology is with the underlying geometry≫. This statement to a certain extent can be also considered misleading, because, as said above, GRPs can be unsuccessful[4] and a consistent part of nonadjacent node pairs can be neglected in the computation of S2 and S3 measures, therefore these two measures are not apt to evaluate congruence in complex networks in general, and might fail also in hyperbolic networks (see Supplementary Note 1 for details). Indeed, to be fair, Krioukov et al.[4] never claimed in their study that S2 and S3 are designed to be measures of network congruence, but they implied that, to a certain extent and for what regards only the part of topology that is explored by successful GRPs, S2 and S3 can offer approximated information on the congruence of the GRPs and the associated GSPs with the underlying geometry. Yet, a well-posed measure that exactly evaluates the congruence of a network topology with the underlying geometry should summarize the deviation of the projections of all the topological shortest paths (pTSPs) with respect to a geometrical reference, as we explain providing references in the Supplementary Note 1. Indeed, we remind that, differently for the topological shortest paths' measures, the GRP: (i) is directed, (ii) can be asymmetric and (iii) can unsuccessfully stop before arriving to destination. These features are not appropriate to offer an exact and robust measure of congruence in complex networks.

We believe that research needs to progress forward "standing on the shoulders of giants" such as Krioukov and colleagues[4]. Indeed, we offered enough evidences to support the need to design: (i) navigability measures that are precise because account for all the GRPs

(successful and unsuccessful); (ii) congruence measures that are general (work with any type of underlying geometry) and are exact (account for all the network topology).

To address the first point, a measure of navigability that accounts for all GRPs (successful and unsuccessful) was already proposed by our group in the studies of Muscoloni and Cannistraci[17,18]. The name of this measure is GR-score or GR-efficiency (GRE) and it allows assessment of GR navigability with a unique score that integrates both the concepts of success ratio and geometrical stretch: a path is assigned a GR-score of 0 if unsuccessful (worst case), a GR-score of ]0,1[ if successful with a stretch greater than 1, and a GR-score of 1 if successful with a stretch 1 (best case). Most importantly, as shown in Fig. 1 of Muscoloni and Cannistraci[17], the introduction of the GRE provides a unique solution when success ratio and hyperbolic stretch suggest conflicting results. The mathematic definition and details of GRE are provided in a dedicated "Methods" section.

To address the second point, a measure of congruence is presented in the next section. This was not possible to be achieved in 2010 in the study of Krioukov et al.[4], because it was not available an efficient algorithm to detect, enumerate and measure the projections of all the possible shortest paths between nonadjacent pairs of nodes in a network, which is a demanding computational problem. In our study we present an efficient algorithm to address this computational problem in large networks, therefore we can propose a measure of congruence that is general (not only for hyperbolic networks) and exact (because it considers all possible shortest paths).

## The geometrical congruence problem

Up today, in network geometry, the notion of congruence was visually and qualitatively introduced when the projections of the topological shortest paths (pTSP) follow closely their associated geodesic in the underlying space[1,4]. However, as we stressed in the introduction and in the previous section on greedy navigability, a quantitative computational measure of geometrical congruence is missing in network science, and this study aims to fill this conceptual gap.

A first concern to address is on the use of the congruence term. In mathematics, the ordinary geometrical sense of this word means that two objects, when they are superimposed (possibly after a mirroring operation), are perfectly aligned. In this respect two geometric objects are either congruent or not, and therefore the attribute associated with the term congruence in mathematics is a binary quantity: presence or absence. When this condition is satisfied, we will call it explicitly hard congruence. Hard in the sense of binary condition because the etymological meaning of congruence does not imply this hard interpretation. Indeed, the term congruence comes from Latin congrŭens -entis and it is believed to be composed of two words cum (together) and (g)ruēre (to move/to come fast; whereas the g is a guttural sound that has the intention to reinforce). Therefore, congruence reads as "to come very fast together", which can be used not necessarily in a binary manner but also in a relaxed form, in order to express in general the extent to which the trajectory of two paths run close together.

A second concern, evident in the example of Fig. 1 and addressed in Fig. 2, is that between each pair of nonadjacent nodes there is only one geodesic but possibly multiple shortest paths. In addition, all nonadjacent node pairs should be taken into consideration to express a collective behavior. Therefore, the hard congruence should be defined in a statistical manner as the extent to which the difference between the distribution of average shortest paths projections and the distribution of the geodesics in a network is not statistically significant.

Finally, the third concern is that evaluating the congruence of a network with its associated geometry is a general problem and not specific to a certain type of geometry. Therefore, we aim to propose a solution that is general too. However, without loss of generality, here we will discuss the paradigmatic case of hyperbolic networks since

they are related with the topic of network navigability, as mentioned in the introduction.

Figure 1 represents many features of a hyperbolic network with $\gamma = 2.5$ generated with the nPSO model (parameters details in Fig. 1 legend). The first conceptual dissonance emerging from Fig. 1e, f is that for a certain geodesic we can have multiple topological shortest paths that visibly diverge from the geodesic, therefore a first step is to analyze the distribution of this divergence for all nonadjacent node pairs in the network. Figure 1d reports the histogram with the distribution of the ratio between the hyperbolic geodesic (GEO) and the average pTSP ($\overline{pTSP}$) in the network. We remind that the current accepted belief is that, in hyperbolic networks with $\gamma < 3$, the pTSPs follow closely their associated hyperbolic geodesics in the underlying space[1,4] and, since this is defined as geometrical congruence of the network topology with the underlying geometry, hyperbolic networks are believed geometrically congruent. However, if this were true, we should expect that in this network with $\gamma = 2.5$ (in Fig. 1d) the distribution of the ratio $GEO/\overline{pTSP}$ would be concentrated around one. Unlikely, this is not the case. Indeed, Fig. 1e, f report that the mean and mode scenarios are very far from the one theoretically proposed, and a rare scenario such as the one in Fig. 1g is close but not convincingly matching the theoretical conclusion. A possible reason for this discrepancy could be that due to finite size effect[19] the geometrical congruence might emerge in the asymptotic regime for growing node size N of the network, and one of the objectives of this study will be to leverage computational tools to investigate the extent to which this hypothesis is valid for growing node size N.

The second step is to move forward with a statistical analysis to appraise whether the distribution of geodesics between all pairwise nonadjacent nodes significantly differ from the distribution of the $\overline{pTSP}$ between the same nodes. Note that $\overline{pTSP}$ is the mean of the pTSP between two nodes, we take the mean as centrality measure because for each geodesic there might be multiple pTSP. Figure 2 displays a comparison of these two distributions when we fix a parameter of the nPSO model and we vary the others. This helps to discuss how the congruence of geodesics and pTSP varies according to a certain specific structural property which is adopted to shape the hyperbolic network according to the nPSO model. Three structural properties are discussed: if average degree $\bar{d}$ grows (m is the direct nPSO model parameter, and $\bar{d}$ is about 2*m for sparse networks; we consider m = [2, 6, 10] hence $\bar{d}$ about [4, 12, 20]), network density increases; if temperature $T$ grows (we consider $T = [0.1, 0.5, 0.9]$), clustering decreases; if $\gamma$ grows (we consider $\gamma = [2, 2.5, 3]$), super-hub structure of the network is mitigated. We keep $N = 100$ because this network node size is enough to properly discuss these structural properties. For each panel of the figure the solid line indicates that community organization is not imposed ($C = 0$, this is equivalent to the classical uniform PSO model[11]) and the dashed line indicates that networks with 4 communities are considered ($C = 4$). In general, these two different community organizations seem not relevant for our investigation and therefore they are not further discussed. Figure 2a has three panels where from left to right the average degree increases while the other parameters are fixed to their intermediate value. It emerges that when average degree grows, $\overline{pTSP}$ between nodes shrinks and approximates better the geodesic whose distribution is unimodal. The distribution of $\overline{pTSP}$ is multimodal because each peak is associated with a different possible topological shortest path length. For small average degree equal to 4 the network is closer to a tree and the peaks of the distribution are associated with topological shortest paths of lengths 2, 3, and 4, respectively, with path 3 being the prevalent. When average degree is 12, the $\overline{pTSP}$ distribution has two peaks that are associated with TSP of lengths 2 and 3. When average degree is 20, there is a prevalence of TSP of length 2 because the network is quite dense. The peak for topological shortest paths of length 4 are present only in Fig. 2a because that is the only subplot with parameter $d = 4$, the network is

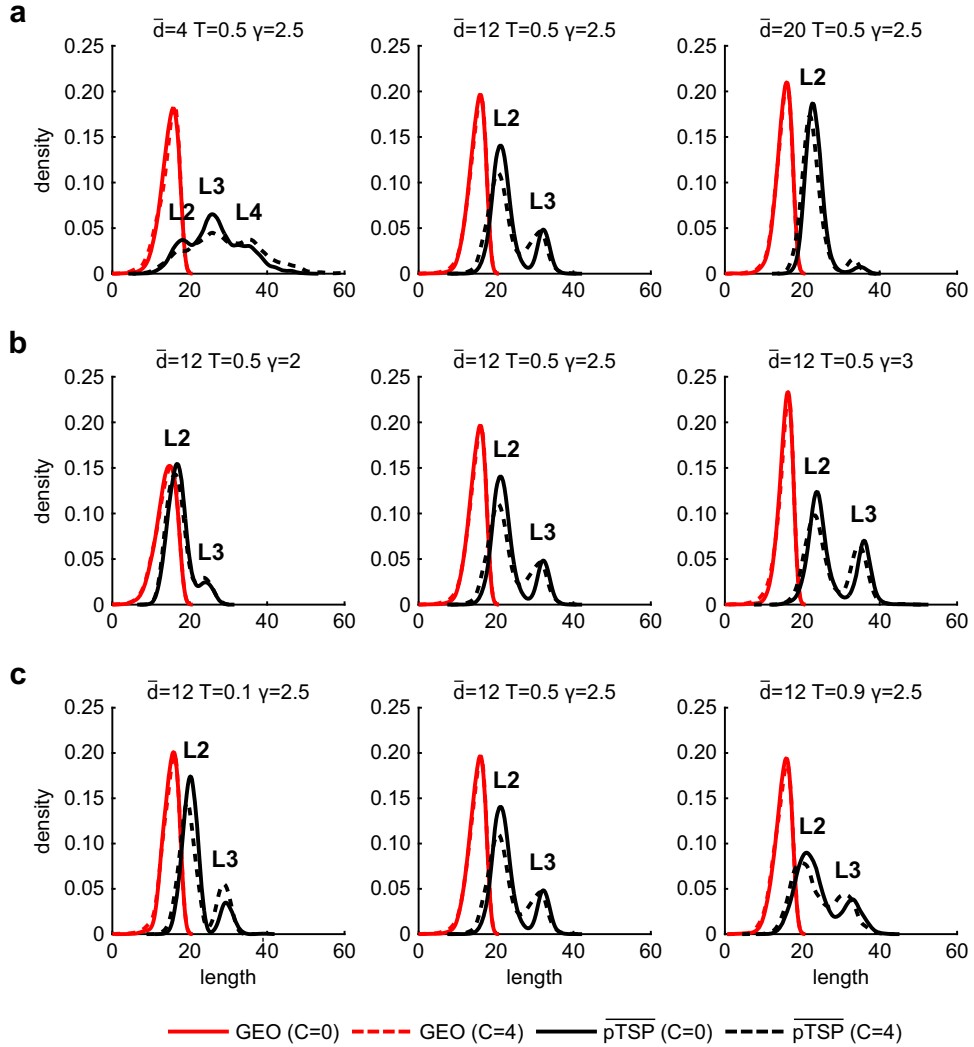

**Fig. 2 | Comparison of GEO and $\overline{pTSP}$ distributions in hyperbolic networks.** For each subplot, the nPSO network has been generated with parameters $N = 100$, $C = [0, 4]$ and values of $\bar{d}$, $T$, γ as indicated on top of each subplot. In particular: (**a**) we fixed $T = 0.5$, γ = 2.5 and varied $\bar{d} = [4, 12, 20]$; (**b**) we fixed $\bar{d} = 12$, $T = 0.5$ and varied γ = [2, 2.5, 3]; (**c**) we fixed $\bar{d} = 12$, γ = 2.5 and varied $T = [0.1, 0.5, 0.9]$. The different results for $C = 0$ and $C = 4$ are shown within each subplot with a solid and dashed line, respectively. For a given network, we have computed the geodesics (GEO) and the $\overline{pTSP}$ for all pairs of nonadjacent nodes. Then, for both GEO and $\overline{pTSP}$ we have estimated by kernel-density the probability density function, which is reported in the subplots (GEO in red, $\overline{pTSP}$ in black): $x$-axis represents the length of the path (GEO or $\overline{pTSP}$) and $y$-axis the density function. For $\overline{pTSP}$ we also highlight the peaks of the distribution that correspond to topological shortest paths of lengths 2, 3, and 4.

very sparse, and therefore the path lengths are longer. In Supplementary Fig. 1 we provide further evidence about this phenomenon induced by sparsity. Figure 2b has three panels where from left to right the power-law exponent γ increases while the other parameters are fixed to their intermediate value. It emerges that when γ grows, $\overline{pTSP}$ distribution remarkably diverges from the geodesic. For γ = 2, $\overline{pTSP}$ distribution has a predominance of TSP of length 2 that are also very close to the geodesic. Whereas for γ = 2.5 and γ = 3 the trend is similar and consists of a bimodal distribution with a first peak for TSP of length 2 and a second for TSP of length 3, but in general these paths are less congruent with the geodesic. Figure 2c has three panels where from left to right the temperature $T$ increases while the other parameters are fixed to their intermediate value. It emerges that when $T$ grows (it means that clustering decreases), $\overline{pTSP}$ distribution seems to maintain the same level of divergence from the geodesic distribution. This might mislead to the conclusion that clustering does not seem to impact the geometrical congruence between $\overline{pTSP}$ and geodesic. However, a closer investigation of Fig. 2c suggests that when T grows, $\overline{pTSP}$ distribution kurtosis is modified. This implies that in nPSO

hyperbolic networks with higher clustering the $\overline{pTSP}$ values related to TSP of the same length are more congruent between them. Finally, we perform for each of the 9 subplots of Fig. 2 a Mann–Whitney statistical test to assess in which of these scenarios the geodesic distribution and $\overline{pTSP}$ distribution do not differ and we can accept the hypothesis of hard congruence between geodesics and associated $\overline{pTSP}$. Astonishingly, the result of the statistical test (considering $p$-value < 0.01 as significance level) is that the hypothesis of hard congruence should be always rejected: geodesic and $\overline{pTSP}$ significantly differ for all possible investigated parameter combinations of the nPSO model hyperbolic networks. Therefore, the first finding of this study is that we cannot statistically accept that hyperbolic networks in general are congruent (or hardly congruent, as we suggest in this study) with their underlying geometry, which is the current credence in the scientific literature.

## A mathematical expression to measure geometrical congruence in networks

At this point of our study, we have to raise the level of the scientific precision adopted to investigate the hypothesis of geometrical

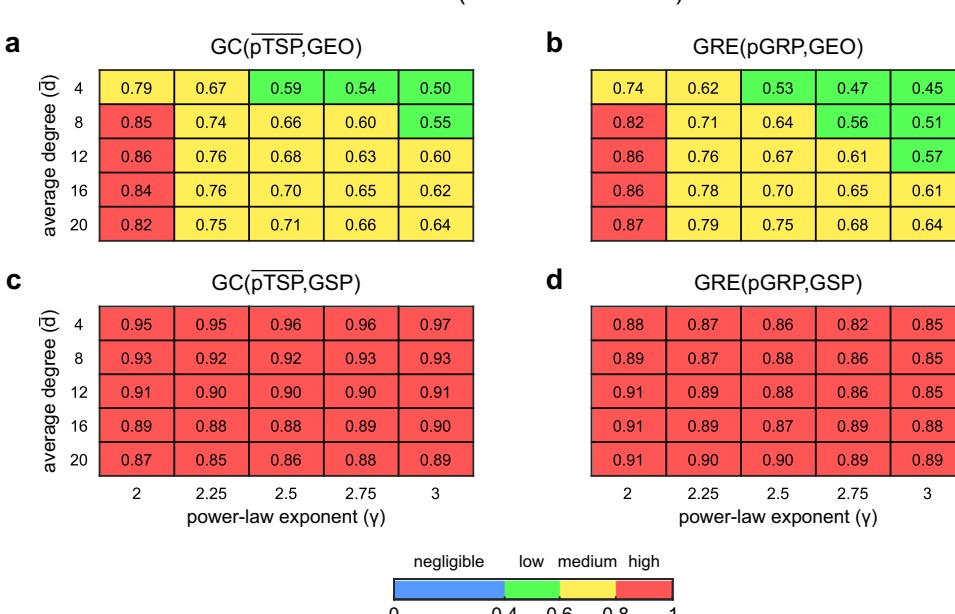

**Fig. 3 | GC and GRE evaluation on hyperbolic networks.** nPSO networks have been generated with parameters $N = 100$, $\bar{d} = [4, 8, 12, 16, 20]$, $T = 0.1$, γ = [2, 2.25, 2.5, 2.75, 3] and $C = 4$. For each combination of parameters, 10 networks have been generated. For each network we have computed: (**a**) GC($\overline{\text{pTSP}}$, GEO), (**b**) GRE(pGRP, GEO), (**c**) GC($\overline{\text{pTSP}}$, GSP) and (**d**) GRE(pGRP, GSP). Each heatmap reports the mean value (over 10 network realizations) of the respective network measure for each combination of $\bar{d}$ and γ in the nPSO generative model.

congruence in network geometry (which returns particularly useful for hyperbolic networks), and to do so we have to invent network science tools that allow to dive deeper in the conceptual and methodical definition of geometrical congruence. We let you notice that current network science literature discusses of proximity and congruence of the geodesic to the pTSP in a qualitative and visual-based fashion that might leave space to misinterpretation and misunderstanding. Hence, the second achievement of this study is a methodological contribution: we introduce a general measure of *geometrical congruence* in complex networks that will be fundamental to quantitatively evaluate the extent to which geometrical networks are congruent with their underlying geometry. As we remarked in the previous section, here the concept of congruence is used to express the extent to which the trajectory of two paths run close together, which is different from the concept of hard congruence standardly adopted in mathematics.

Given a network with $n$ nodes and $e$ edges, we define the geometrical congruence (GC) as

$$\text{GC}(\overline{\text{pTSP}}, \text{RD}) = \left( \frac{2}{n \cdot (n-1) - 2 \cdot e} \right) \cdot \sum_{i<j} \frac{\text{RD}(i,j)}{\overline{\text{pTSP}}(i,j)} ; \text{with} (i,j) \in \widetilde{E}$$

where $\widetilde{E}$ is the set of pairs $(i, j)$ of nonadjacent nodes.

The computation of the $\overline{\text{pTSP}}(i,j)$ is nontrivial because it requires to find all the possible TSP between $(i,j)$. The technical details on how to design algorithms to address this challenging computational problem are provided in a dedicated section below. RD$(i, j)$ can be any node pairwise reference distance (not necessarily restricted to the geodesic) that is associated with the geometry. For instance, in this study we consider RD$(i, j)$ equal to the geodesic (GEO) in one case and to the geometrical shortest path (GSP) in the second case. Hence, in the first case we measure the GC with the geodesic, in the second case we measure the GC with the GSP. Since a GC = 0.5 means that on average in the network $\overline{\text{pTSP}}(i,j)$ is twice the length of RD(i,j) (indicating a low congruence) we consider the following definition of the scale of values for GC: GC = [0, 0.4 [indicates negligible congruence; GC = [0.4, 0.6 [indicates low congruence; GC = [0.6, 0.8[indicates

medium congruence; GC = [0.8, 1] indicates high congruence. A basic requirement of previous measures based on navigability is to have a connected network (or to consider the largest connected component), since it is meaningful to measure the navigability only between connected parts. The same basic requirement applies also to the measure of geometrical congruence. Finally, when the only available knowledge is the unweighted network topology, in order to determine the links weights for computing the GSP to adopt as RD, according to Muscoloni et al.[18] a dissimilarity measure between adjacent node pairs associated with the underlying geometry such as the repulsion-attraction rule (RA) or the edge betweenness centrality (EBC) can be used.

**Geometrical congruence in networks with patent geometry: the case of hyperbolic networks**

Figure 3 reports the values of GC measure and greedy navigability measure in hyperbolic networks generated with the nPSO model across different parameter combinations. In particular, Fig. 3a shows a heatmap with average GC($\overline{\text{pTSP}}$, GEO) on 10 realizations of nPSO hyperbolic networks ($N = 100$, $T = 0.1$, 4 communities) spanned across a large combination of average degree $\bar{d}$ and power-law exponent γ. From Fig. 3a emerges that high GC($\overline{\text{pTSP}}$, GEO) is reached in these hyperbolic networks only for γ = 2 and, most importantly, the measure of GC($\overline{\text{pTSP}}$, GEO) is strongly matched with a measure of navigability (Fig. 3b) termed greedy routing efficiency[17,18] (GRE, whose range of values is between 0 and 1, see "Methods" for details). We computed GRE(pGRP, GEO) by comparing the projection of the greedy routing paths (pGRP) with the respective geodesics between pairs of non-adjacent nodes. For $T = 0.3$ (see Supplementary Figs. 2–5) the nPSO model networks seem to retain similar congruence and navigability, however for $T = 0.5$ when the clustering vanishes (and consequently hyperbolic geometry vanishes[4,9,11]) also the congruence and navigability are significantly affected, and this result is in accordance with previous conclusions[4]. Figure 3a, b results are confirmed also for networks with $N = 1000$ and no fixed community organization (see Supplementary Figs. 2–5). Hence, the second finding of this study is that in general for γ = [2, 3] the hyperbolic networks generated with the nPSO

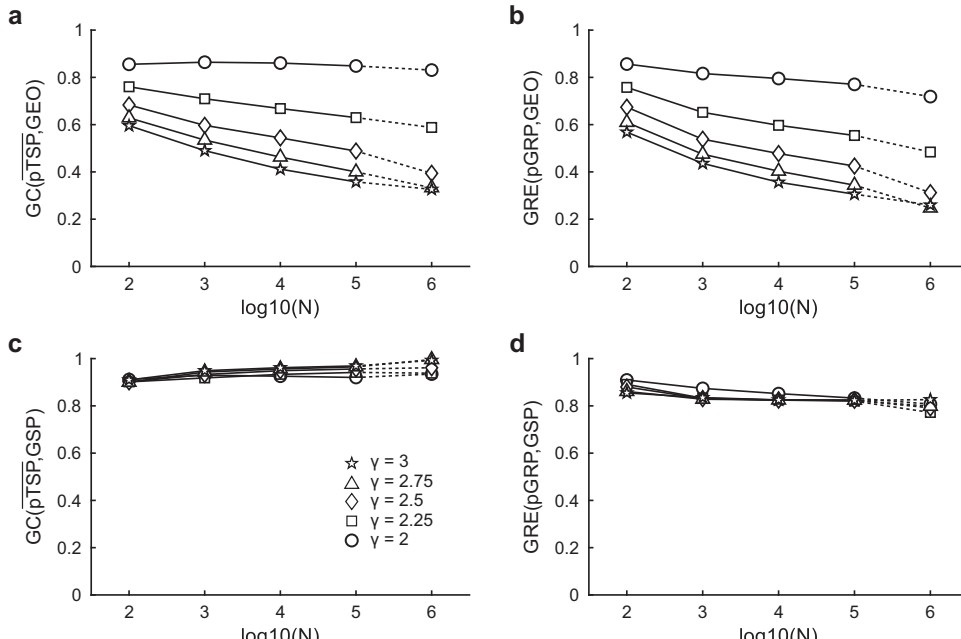

**Fig. 4 | GC and GRE evaluation for increasing network size.** nPSO networks have been generated with parameters $N = [100, 1000, 10000, 100000]$, $\bar{d} = 12$, $T = 0.1$, $\gamma = [2, 2.25, 2.5, 2.75, 3]$ and $C = 4$. For each combination of parameters, 10 networks have been generated. For each network we have computed: (**a**) GC($\overline{\text{pTSP}}$, GEO), (**b**) GRE(pGRP, GEO), (**c**) GC($\overline{\text{pTSP}}$, GSP), (**d**) GRE(pGRP, GSP). Each plot reports 5 curves for the different values of $\gamma$, indicating the mean value (over 10 network realizations) of the respective network measure for increasing network size. Error bars are not reported since negligible. In order to provide an estimation for networks of size $N = 1,000,000$, for each curve we used the spline algorithm to perform interpolation at $N = [100, 1000, 10,000, 100,000]$ and extrapolation at $N = 1,000,000$, the resulting values are highlighted with dashed lines.

model present medium to low congruence and greedy navigability, and that high congruence/navigability (but not hard congruence) emerges only for $\gamma$ proximal to 2 (at least in these nPSO hyperbolic networks). These computational findings are relevant because they significantly correct and refine the results of previous theoretical studies[4,8], which are then included in a review study on network geometry[1], according to which the greedy navigation in hyperbolic networks with $\gamma < 3$ can always find ultrashort paths which follow the geodesics, and thus navigation in these hyperbolic networks is maximally efficient. This is not true for all hyperbolic networks, and in Fig. 3a, b (and Supplementary Figs. 2–5) we offer computational evidence based on GC and GRE that this is not true for PSO[11] and nPSO[9] hyperbolic networks. A possible objection to our result could be that the geometrical congruence might emerge in the asymptotic regime for growing node size N of the network and that in this study we offer evidence only till $N = 1000$. However, addressing this concern is nontrivial because the computational complexity of the algorithm is onerously growing with N and $\gamma$. Therefore, to investigate this concern we had to develop an optimized and parallelized algorithm that is discussed in the next section. After months of computing we obtained results to investigate how the trend of GC($\overline{\text{pTSP}}$, GEO) (Fig. 4a) and GRE(pGRP, GEO) (Fig. 4b) varies for growing N (till $10^5$ with computation and $10^6$ with extrapolation, see Fig. 4 legend for details) and $\gamma = [2, 2.25, 2.5, 2.75, 3]$. Since the computation is burdensome in time, the other parameters that are less important for our investigation were fixed to reference values: $\bar{d} = 12$, $T = 0.1$, and $C = 4$. More parameter combinations for networks of $N = 10^4$ are provided in Supplementary Figs. 6, 7. The results of this computation confirm that the trend of GC($\overline{\text{pTSP}}$, GEO) (Fig. 4a) and GRE(pGRP, GEO) (Fig. 4b) is not growing with node size N. Therefore, the results we present discourage the notion (included in a review study on network geometry[1]) that in the asymptotic regime the greedy navigation in hyperbolic networks with $\gamma < 3$ can always find ultrashort paths which are (hardly) congruent with the geodesics. This is the third finding of our study. Interestingly, if we

change the reference distance from geodesic to GSP (the green line in Fig. 1b) then the measures GC($\overline{\text{pTSP}}$, GSP) and GRE(pGRP, GSP) show high congruence and high navigability for any parameter combination (Fig. 3c, d) at $T = 0.1$. And this is confirmed in Fig. 4c, d also for growing N. On one side, this result suggests that using the GSP as reference distance for GC and GRE is misleading if the goal is to investigate the geometrical congruence of a network with its underlying geometry for different $\gamma$ values, in which case the geodesic should be used as reference. On the other side, the proposed GC measure seems properly designed, since it is strongly associated with GRE on hyperbolic networks with high clustering ($T = 0.1$) and, as expected according to theory[4], when temperature increases ($T = 0.3$ and $T = 0.5$, see Supplementary Figs. 2–7), clustering decreases, hyperbolic geometry tend to vanish and GC($\overline{\text{pTSP}}$, GSP) has a tendency to deviate from GRE(pGRP, GSP) which becomes more evident for $C = 4$ and $N = 1000$ (Supplementary Fig. 5). Therefore GC($\overline{\text{pTSP}}$, GSP) can still be considered an interesting marker to compare differences of geometrical congruence between structural and weighted connectivity across complex networks with latent geometry such as brain connectomes as we will discuss below in the last section of the Results.

**An algorithm to compute geometrical congruence in networks**

The essence of this section is on how to design an optimized algorithm that reduces 26 years of worst scenario computation to one week, and this represents the fourth achievement of our study. Indeed, the $\overline{\text{pTSP}}$ is the average projection of the topological shortest paths between two nonadjacent nodes in the network. Therefore, in order to compute the $\overline{\text{pTSP}}$, we need to find all the topological shortest paths for each nonadjacent node pair in the network that is a nontrivial computational task dependent on different topological features.

A standard solution to this problem is given by a recursive approach (the reasons to prefer a recursive approach to others are discussed in the respective "Method" section). Starting from a source node and performing a recursive visit to the neighbors (while avoiding

loops) up to a recursion depth equal to the maximum shortest path of the source node, it is guaranteed to traverse once and only once all the topological shortest paths from the source node to all the other nodes. The projection (geometrical length) of such paths can be stored while traversing them, in order to obtain the $\overline{\text{pTSP}}$ from the source node to all the others at the end of the recursion process. If the recursion procedure is repeated starting from each node as source, the result is the $\overline{\text{pTSP}}$ between all node pairs. Note that the recursions starting from different source nodes are independent, therefore they can be computed in parallel. We will refer to this algorithmic solution as *brute-force* variant, since for each source node the recursion depth is set to its maximum shortest path. However, one of the main drawbacks of the brute force algorithm is the redundancy in computing some shortest paths multiple times. To address this issue, we propose an optimized algorithm that reduces the redundancy of the computation preserving the parallelization. Our optimized strategy is twofold and here we highlight the two main aspects of this algorithm.

The first aspect on which is based the optimization algorithm is a strategy that avoids processing the same input multiple times: given an arbitrary node sequence, and before to enter in the parallelized recursions, the algorithm pre-assigns to each node a maximum recursion depth adjusted neglecting from the computation of the maximum the previous nodes in the sequence. This, as a matter of fact, will exploit the symmetry of the system avoiding that the maximum recursion depth of a node might be conditioned by the recursion depth of previous nodes in the sequence, which will be independently computed in parallel. In simple words, if a node i is before a node j in the sequence and i determines the maximum recursion depth of j, then the computation to find all shortest paths from i to j will not be repeated from j to i. This is the first key contribution of our optimized algorithmic, which is termed *memoization*. The term memoization has a very specific meaning in computer science that we discuss in depth in the respective method section associated with this algorithm. For non-technical readers, this key contribution can be viewed as a memorization strategy ensuring that a function does not run for the same inputs more than once, by keeping a record (memory) of the results for the given inputs.

The second aspect on which is based the optimization algorithm is a strategy to produce a node sequence that improves the efficiency of the memoization part. Any permutation of the node sequence would provide the same results of the $\overline{\text{pTSP}}$ between node pairs, but different permutations can imply different maximum recursion depths for each node in the sequence, therefore different computational times. In particular, the more the first nodes in the sequence have many topological shortest paths of high length, the higher the likelihood that the following nodes will have a decreased recursion depth, since those long paths will be removed from the computation of the maximum. Therefore, a criterion to generate a node sequence that allows a computational time reduction is crucial. Theoretically, one could define an objective function (representing the computational load) as function of the maximum recursion depths of the nodes, then find the exact sequence of nodes that minimizes it. However, considering that there are $N!$ permutations of the nodes to test, this solution is unfeasible in practice using classical von Neumann computing architecture. To address this issue, we propose a heuristic approach that provides an effective solution and consists in ordering the nodes by decreasing average topological shortest path, which can be computed efficiently and guarantees that the nodes with many long paths are positioned first. This is the second contribution in our algorithmic solution, which is termed *prioritization*. We note that choosing an optimized order of the source nodes (prioritization) is strictly synergetic with the memoization, whereas it would not have any effect on the brute-force approach. Indeed, the prioritization step is antecedent to the memoization, which exploits the node sequence ordering provided by the prioritization.

Furthermore, we clarify that the prioritization and memoization are performed in advance before to start any recursion. This keeps the recursions of different source nodes independent allowing them to process in parallel. For a careful understanding and discussion of the technical details behind the algorithm design we refer to: Method section (Computation of the $\overline{\text{pTSP}}$ between all node pairs: algorithm design), Supplementary Note 2 section (Pseudocode to compute the $\overline{\text{pTSP}}$ between all node pairs), Supplementary Note 3 "Space and time complexity", Supplementary Note 4 "Running time estimation".

After introducing the basic concepts behind the two types of algorithms (brute-force and our optimized algorithm), we will move forward discussing their running time and time complexity. In Fig. 5a we report the running time (average time on 10 networks realizations by nPSO model) of the two different network congruence algorithms using parallel computation (128 cores). Each of the five curves indicates networks with five different values of γ (2, 2.25, 2.5, 2.75, 3) and are obtained for increasing network node size values ($N = 100$, 1 K, 10 K, 100 K). Average degree, temperature and number of communities in the nPSO are respectively fixed to $\bar{d} = 12$, $T = 0.1$ and $C = 4$, because from the previous section we noticed that these are parameters of second relevance for our investigation and that we should concentrate our attention on understanding the impact of $N$ and γ. The running time of our optimized algorithm (Fig. 5a, red curve) is negligible (one second) till $N \sim 1000$, it is small for $N \sim 10$ K (few minutes), whereas it becomes impactful for $N \sim 100$ K (in the order of days), although still feasible as a large-scale computation. In comparison, for $N \sim 100$ K the running time of brute-force (Fig. 5a, black curve) significantly increases to more than 2 months of computation (see Supplementary Note 4 for details on the running time estimation) for the highest values of γ (2.75 and 3). Hence, the second step of our investigation is to deepen our understanding of the computational processes behind the algorithmic running time differences observed in Fig. 5a for $N = 100$ K and the highest γ values. To this aim in Fig. 5b we repeat the same analysis of Fig. 5a but specifying the difference between results obtained with and without parallel computation, on networks of size $N = 100$ K and for γ = [2, 2.5, 3]. The first evidence is that both on single core (Fig. 5b right panel) and 128 cores parallelization (Fig. 5b, left panel), regardless of the γ value, our algorithm design leads to a reduction of around 90% with respect to the computational load needed by brute-force. In addition, it is evident the impact of the power-law exponent γ on the running time. In case of γ = 3, we notice the reduction from a worst-case scenario of 26 years for brute-force on single core to the actual scenario of 7 days for our algorithm on 128 cores, representing a great computational achievement, which remains relevant even when the brute-force algorithm is parallelized on 128 cores (2.4 months running time). Therefore, the third step of our investigation aims to address two questions: to understand the reason behind this remarkable running time reduction gained by our optimized algorithm; to identify the reason behind the impactful running time increase occurred for γ = 2.75, 3 regardless of the algorithm design.

In order to address these two questions, we have to discuss the time complexity. The general time complexity of this class of algorithms (both brute-force and our optimized algorithm) is analyzed in details in a dedicated section in Supplementary Note 3 and, with some approximations, can be formulated as

$$O\left( N \cdot \sum_{L=2}^{d} M_L \cdot k^L \right)$$

where $k$ is the average node degree, $d$ is the network diameter, $M_L$ is a coefficient that expresses the proportion of nodes with recursion depth equal to $L$. The analysis of this formulation is not trivial, since it depends on the distribution of the recursion depths, which in turn are affected by different topological features of the network. One of such features is certainly the average node degree (or similarly the edge

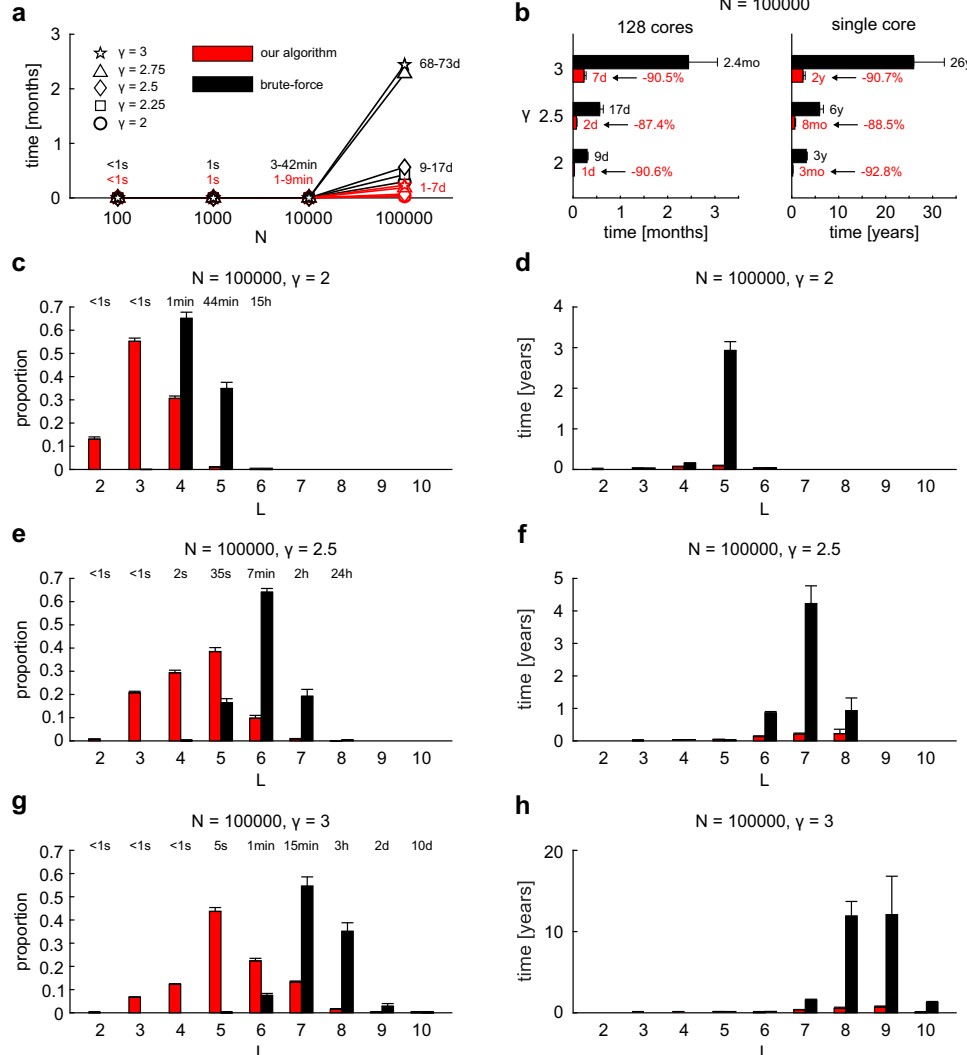

**Fig. 5 | Running time estimation.** We generated nPSO networks with parameters $N = [100, 1000, 10,000, 100,000]$, $\bar{d} = 12$, $T = 0.1$, $\gamma = [2, 2.25, 2.5, 2.75, 3]$ and $C = 4$. **a** The panel reports the running time over increasing network size, both for our algorithm (red) and brute-force (black) using parallel computation, as separate curves for different values of $\gamma$. **b** The panel reports, for $\gamma = [2, 2.5, 3]$, the estimated running time with and without parallel computation, both for our algorithm and brute-force, on networks of size $N = 100,000$. The percent of decrease is shown. **c**, **e**, **g** The panels report, for $\gamma = [2, 2.5, 3]$ respectively, the proportion of nodes associated with each recursion depth, both for our algorithm and brute-force, on networks of size N = 100,000. In addition, on top of the barplots, we report the mean running time required to reach each recursion depth. **d**, **f**, **h** The panels report, for $\gamma = [2, 2.5, 3]$ respectively, the running time that would be required to compute the recursive section of the algorithm for each recursion depth and without considering parallel computation, both for our algorithm and brute-force, on networks of size $N = 100,000$. In all panels, values are averaged over 10 network realizations and error bars are shown, except for panel (**a**) due to visual overlap with the symbols. For more details, please refer to the dedicated section Supplementary Note 4.

density). The higher the average node degree, the shorter will be the topological shortest paths, the lower will be the recursion depths. In the case of networks with a power-law degree distribution, such as the nPSO networks analyzed, the power-law exponent γ also affects the recursion depths distribution. Indeed, the lower the γ, the higher the presence of hubs connecting most of the network, making the topological shortest paths (and in turn the recursion depths) shorter. Indeed, networks with γ < 3 are defined as ultra-small world[7].

To address the first question on the reason behind the large running time improvement of our optimized algorithm we note that the distribution of $M_L$ (Fig. 5c, e, g), which is the proportion of nodes with recursion depth equal to L, is significantly shifted towards small L values for our optimized algorithm (red bars Fig. 5c, e, g) in comparison to the brute-force (black bars Fig. 5c, e, g). The fact that this attenuation happens similarly for any value of power-law exponent γ can explain why in Fig. 5b our optimized algorithm displays a similar reduction of −90% in running time regardless of γ. From the theoretical

point of view, recovering the time complexity formula provided above, this implies that our optimized algorithm attenuates the cost of large $L$ values and this seems the main reason for its significant running time reduction. This is confirmed by the fact that the dramatic running time (without parallel computation, Fig. 5d, f, h) of the brute-force algorithm (black bars Fig. 5d, f, h) for large $L$ values is significantly attenuated by our optimized algorithm (red bars Fig. 5d, f, h), and this happens likewise for any value of power-law exponent γ.

To address the second question on the reason behind the impactful running time increase occurred for γ = 2.75, 3 regardless of the algorithm design, we note that with our optimized algorithm for γ = 2 (red bars Fig. 5c) most of the recursion depths are distributed between 2 and 4 (average 3.2), with a network diameter of 6. For γ = 2.5 (red bars Fig. 5e), most of recursion depths are distributed between 3 and 6 (average 4.4), with a diameter of 8. Finally, for γ = 3 (red bars Fig. 5g), most of the recursion depths are distributed between 3 and 7 (average 5.3), with a diameter of 10. If we now look at the brute-force,

we note that for γ = 2 (black bars Fig. 5c) most of recursion depths are between 4 and 5 (average 4.3), for γ = 2.5 (black bars Fig. 5e) between 5 and 7 (average 6), for γ = 3 (black bars Fig. 5g) between 6 and 8 (average 7.3). In summary, the mean recursion depths of both algorithms, as well as their distributions, clearly shift towards larger L values for higher γ, with the brute-force algorithm which is severely afflicted because it shifts 1–2 links more than our optimized algorithm. This is, regardless of the algorithm, at the origin of the strong time complexity increase especially for γ = 2.75, 3, and it is reflected also in the running time (without parallel computation) reported in Fig. 5d, f, h, where looking at the brute-force algorithm from γ = 2 to γ = 2.5 the time remains under 5 years, but from γ = 2.5 to γ = 3 the time increases to values that are in the scale of decades.

### Geometrical congruence in networks with latent geometry: the case of brain connectomes

Finally, in the last section of this study, we present results on how to apply the measures of greedy navigability and geometrical congruence on networks with latent geometry. A network is defined with latent geometry when we do not know the coordinates of the nodes in the original space that generated the network topology. Indeed, if we have the network connectivity and the coordinates of the nodes we can infer what type of geometrical model fits the network geometry.

Contrary to the greedy navigability measures that can be applied only on networks with patent geometrical space, the proposed geometrical congruence measure can be employed as a marker to reveal differences in real networks with latent geometrical space. In simple terms this means that to compute the greedy navigability measures we need to know or hypothesize: network connectivity and nodes geometrical coordinates (known or inferred by network embedding). The node geometrical coordinates are in turn used to compute the node pairs distances considered in the myopic transfer that forms a greedy routing path. As a further clarification, if the space is originally latent and a patent space is inferred using network embedding algorithms[18], this is an intermediate step that is not related with greedy navigability measures, which therefore cannot be directly applied to networks with latent space. Differently, the geometrical congruence can be measured having at hand just connectivity information: the network connectivity and its links weights (node coordinates are not required). In the unwished scenario that the network topology is unweighted, a dissimilarity measure between adjacent node pairwise associated with the underlying geometry such as the repulsion-attraction rule (RA) or the edge betweenness centrality (EBC) can be used[18].

Most real networks are associated with a latent space. It means that even though their nodes have coordinates in a visible 2D or 3D space, this visible space does not match the latent geometrical space according to which the connectivity is shaped. This makes navigability measures difficult to apply because the latent space is often multi-dimensional and the number and type of variables are unknown. For instance, a recent study demonstrated that the structural core of a particular type of maritime networks is associated with variables that measure the international trade statuses of countries[20]. Emblematic examples of latent space networks are social networks, where nodes are individuals and possible latent variables are geographical location (visual space), cultural interests, political interests, economical interests, job, education, different types of hobbies, etc. Another paradigmatic example is macroscale structural MRI brain connectomes, where the nodes are brain areas and the latent variables are: 3D anatomical location (visual space) and many unknown developmental neurobiological variables that determine the circuital maps of the brain and their complex connectivity architecture. Indeed, knowing the 3D location of the nodes is not enough to reconstruct the latent geometrical space to which the connectivity is associated.

In this study, we focus our investigation on the comparison between greedy navigability and geometrical congruence in macroscale structural MRI brain connectomes, aiming to address three questions. The first is inherent to this network geometry study and regards whether we can provide any computational evidence that geometrical congruence, as theoretically expected, offers a concrete advantage on measuring properties of latent space networks with respect to greedy navigability. The second is an open problem in network neuroscience[21]. Several measures of greedy navigability were proposed by Krioukov et al. for hyperbolic networks, such as hyperbolic stretches[4], but they were supposed to work primarily in hyperbolic networks. Unfortunately, we offered evidence that even in hyperbolic networks these measures show some inconsistencies, because of the limitations reported in the Results section of this study that is dedicated to navigability. Therefore, to address such issues, Muscoloni et al.[17,18] designed a measure of network greedy navigability efficiency for complex networks in general (not only hyperbolic), and successively Seguin et al.[21] discussed and motivated its application on brain connectomes. Here we wish to make a step forward to investigate whether greedy navigability efficiency, and consequently also geometrical congruence, can be used as markers to differentiate different phenotypical states in brain connectomes. The third is a relevant debated problem at the interface between network geometry and network neuroscience. It regards whether navigability and congruence can be used to gain any evidence at support of the thesis that macroscale structural MRI brain connectomes might have a latent developmental neurobiological geometry which accounts for more information content than the 3D visible Euclidean geometry. Indeed, brain networks are associated with a developmental neurobiological geometry which also influenced by neuronal plasticity[22] and, according to some studies, might be approximated by hyperbolic distances[12–14]. However, as a matter of fact, it is latent because both the type of geometry and the coordinates of the nodes in this developmental neurobiological space are unknown.

To address these questions, we consider 614 macroscale structural MRI brain connectomes of healthy resting-state individuals from Faskowitz et al.[23], that can be divided for sex (230 male versus 384 females) or for age range (223 in [7 to 30] and 215 in [55 to 85]). Connectomic links can be weighted according to two different dissimilarities: the inverse of the number of streamlines (NOS) and the mere three-dimensional Euclidean (3D) distance between two nodes. These two different dissimilarities are associated with two different geometrical spaces. The NOS is obtained by diffusion tensor imaging tractography analysis and therefore is related with an unknown developmental neurobiological geometry, which is behind the connectivity captured by anatomical imaging. Instead, the mere three-dimensional Euclidean (3D) distance can be considered as a control hypothesis. It can help to evaluate the extent to which, in terms of navigability and congruence, the latent neurobiological-driven geometry encoded in NOS explains brain connectivity differences better than the visible Euclidean geometry.

Results in Fig. 6a, d evidence that GC($\overline{\text{pTSP}}$, GSP) difference (considering NOS) is statistically significant ($p$-value < 0.01, Mann–Whitney test) both for sex and age, respectively. Although this result is significant, it does not say anything about the extent to which GC can be used as a marker that quantifies differences in brain connectomes and whether it works better than the greedy navigability efficiency—also known in network neuroscience as Efficiency Ratio ($E_R$)—proposed in the previous studies[17,18,21]. Therefore, we evaluated the extent to which GC and $E_R$ can discriminate two states considering the area under precision recall curve (AUPR), which is robust against different group sizes[24]. In addition, we evaluated whether a certain value of AUPR is statistically significant in comparison to a random permutation of the labels considering a measure called trustworthiness[25]. We consider AUPR-trustworthiness—with Bonferroni correction over the 4 markers—significant when $p$-value < 0.01. Results in Fig. 6b, c reveal that GC performs always better than $E_R$ to measure the sex

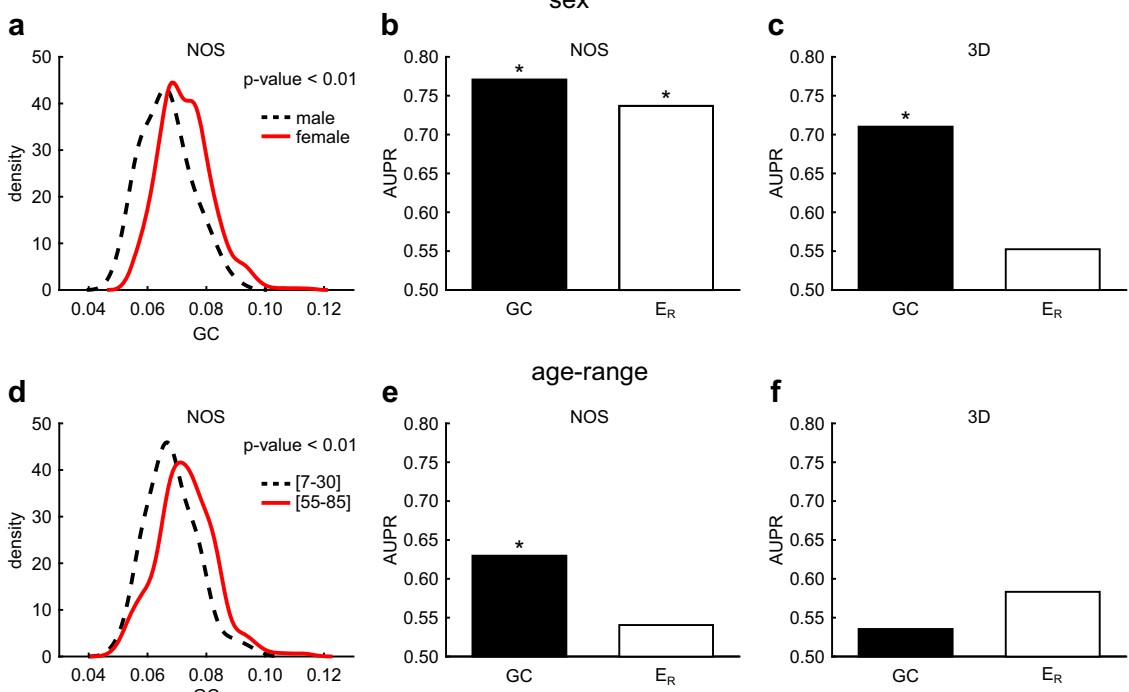

**Fig. 6 | GC as marker for sex and age discrimination in structural connectomes.** We have analyzed a dataset of structural connectomes with sex and age annotation. For each network, we have computed the GC(GSP, $\overline{\text{pTSP}}$) and the $E_R$ with connection weights based on NOS or 3D (see "Methods" for details). **a** We divided the connectomes in two groups related to male and female subjects, for each group we have estimated by kernel-density the probability density function of GC, which is reported in the subplot (male in black dashed line, female in red solid line): *x*-axis represents the GC(GSP, $\overline{\text{pTSP}}$) and *y*-axis the density function. The *p*-value of the Mann–Whitney test shows that GC can significantly discriminate between male and female connectomes (*p*-value < 0.01). **b**, **c** The barplots compare the AUPR of GC and $E_R$ methods in discriminating between male and female connectomes, when using connection weights based on NOS (**b**) or 3D (**c**). An asterisk indicates that the AUPR-trustworthiness[25] with Bonferroni correction over the 4 markers is significant (*p*-value < 0.01). **d** We have repeated the same analysis as in (**a**), but considering two groups of connectomes related to two age-ranges of the subjects: [7–30] (black dashed line) and [55–85] (red solid line). The *p*-value of the Mann–Whitney test shows that GC can significantly discriminate between connectomes of young and elderly subjects (*p*-value < 0.01). **e**, **f** The barplots compare the AUPR of GC and $E_R$ methods in discriminating between connectomes related to two age-ranges, when using connection weights based on NOS (**e**) or 3D (**f**). An asterisk indicates that the AUPR-trustworthiness with Bonferroni correction over the 4 markers is significant (*p*-value < 0.01).

connectomic separation regardless of the type of link weight dissimilarity (NOS or 3D), in addition the AUPR of GC is always statistically significant, whereas the one of $E_R$ is not for 3D Euclidean. Results in Fig. 6e reveal that GC performs better than $E_R$ to measure the age connectomic separation using link weight dissimilarity NOS, and GC is the only one to provide a statistically significant AUPR, whereas $E_R$ does not. Results in Fig. 6f show that both GC and $E_R$ fail, therefore 3D Euclidean weights are not a sufficient source of information to measure values of AUPR that are significant for age. Therefore, the reply to the first question is yes: geometrical congruence, as theoretically expected, offers a concrete advantage on measuring properties of latent space networks with respect to greedy navigability, because the fact that does not need any hypothesis on the node coordinates relaxes a constraint that is source of uncertainty in analysis of latent space networks. The reply to the second question is yes: greedy navigability efficiency, and consequently also geometrical congruence, can be used as markers to differentiate different phenotypical states in brain connectomes. However, we discover that (at least on these data) geometrical congruence works in general better, and this represents the fifth achievement of this study. Finally, all results in Fig. 6 suggest that NOS is a better link weight dissimilarity with respect to 3D. Hence, the reply to the third question is yes: navigability and congruence offer evidence at support of the thesis that macroscale structural MRI brain connectomes might have a latent developmental neurobiological geometry (waiting for more evidences about whether it is hyperbolic[12–14]) which accounts for more information content than the 3D visible Euclidean geometry. However, to avoid misunderstandings,

we clarify that the measures of congruence and navigability discussed in this study are model-free and therefore do not make any assumption on the type of underlying geometry of the complex networks. For this reason, they cannot be directly employed to determine the type of geometry behind complex networks. This might require to build a null geometrical model, to use as a reference, and with respect to which variations of congruence and navigability should be compared. This is not the topic of the present study, and we hope that future studies might investigate how to exploit congruence and navigability to infer the nature (elliptic, Euclidean, hyperbolic) of the latent geometry behind many real networks.

## Discussion

At first, we redefine the concept of congruence in network geometry not as hard congruence, which is the ordinary geometrical sense used in mathematics, but in a relaxed form to expresses in general, and in a not-binary manner, the extent to which the trajectory of two paths run close together. Next, we notice that between each pair of nonadjacent nodes we have only one geodesic but possibly multiple shortest paths. In addition, all nonadjacent node pairs should be taken in consideration to express a collective behavior. Therefore, the hard congruence should be defined in a statistical manner as the extent to which the difference between the distribution of average shortest paths projections and the distribution of the geodesics in a network is not statistically significant. Then, we considered networks with patent (the contrary of latent in *Latin*) geometrical space, meaning that the coordinates of the network nodes in this space are known. Specifically,

we investigate hyperbolic networks generated with the nonuniform popularity-similarity optimization (nPSO) model[9,10]. As a consequence of this investigation, the first finding of this study is that we cannot statistically accept that hyperbolic networks in general are congruent (or hardly congruent, as we suggest in this study) with their underlying geometry, which is the current credence in the scientific literature[1].

The second achievement of this study is a methodological contribution: we introduce a general measure of *geometrical congruence* in complex networks to quantitatively evaluate the extent to which geometrical networks are congruent with their related geometry. Specifically, we defined a measure that considers the average ratio (calculated for each nonadjacent node pair in the network) between a reference value associated with the geometry (for instance the geodesic or any dissimilarity) and the average shortest paths projection.

Performing accurate computation based on this geometrical congruence measure, we could exclude that the congruence between geodesics and average shortest path projections, as well as the associated greedy routing efficiency, are growing with node size N in hyperbolic networks. Therefore, the results we present discourage the notion (included in a review study on network geometry[1]) that in the asymptotic regime the greedy navigation in hyperbolic networks with γ < 3 can always find ultrashort paths which are (hardly) congruent with the geodesics. This is the third finding of our study.

However, in order to compute the average projection of the topological shortest paths between two nonadjacent nodes in the network, we need to find all the topological shortest paths for each nonadjacent node pair in the network, that is a nontrivial computational task dependent on different topological features. Therefore, the fourth achievement of this study was to design an optimized algorithm that reduces 26 years of worst scenario computation to one week using parallel computing.

Finally, differently from greedy navigability measures that need nodes geometrical coordinates (patent geometrical space), we show that the proposed geometrical congruence, as theoretically expected, offers a concrete advantage on measuring properties of latent space networks. The fact that geometrical congruence does not need any hypothesis on the node coordinates relaxes a constraint that is source of uncertainty in analysis of latent space networks. Indeed, we discover that geometrical congruence works in general better than greedy navigability efficiency as marker to differentiate phenotypical states in brain connectomes, and this represents the fifth achievement of this study.

Altogether these findings could have practical impact on real applications for the design and engineering of communications networks, such as Internet, or for quantitative investigation of the topological-geometrical coupling, which is a promising measurable network feature that might be associated with the organization and functionality of brain connectomes or other complex networks[26–29].

## Methods
### Geometrical congruence (GC)
Given a network with $n$ nodes and $e$ edges, we define the geometrical congruence (GC) as

$$\mathrm{GC}\left(\overline{\mathrm{pTSP}}, \mathrm{RD}\right) = \left(\frac{2}{n \cdot (n-1) - 2 \cdot e}\right) \cdot \sum_{i<j} \frac{\mathrm{RD}(i,j)}{\overline{\mathrm{pTSP}}(i,j)}; \text{with} (i,j) \in \widetilde{E}$$

where $\widetilde{E}$ is the set of pairs $(i, j)$ of nonadjacent nodes.

In practice, for each pair $(i, j)$ of nonadjacent nodes we compute the ratio between RD$(i, j)$, which is a reference distance, and $\overline{\mathrm{pTSP}}(i, j)$, which is the mean projection of all the TSP between $(i, j)$. The GC is obtained as the average of such ratios, assuming values between 0 and 1. Note that RD$(i, j)$ = RD$(j,i)$ and $\overline{\mathrm{pTSP}}(i,j)$=$\overline{\mathrm{pTSP}}(j, i)$, therefore we only evaluate for pairs $(i, j)$ such that $i < j$, resulting in a total of $\frac{n \cdot (n-1) - 2 \cdot e}{2}$ undirected node pairs to be averaged. RD$(i, j)$ can be any node pairwise

reference distance. In this study, in one case we consider RD(i,j) equal to the geodesic (GEO) and in another case equal to the geometrical shortest path (GSP).

In the case of GEO, in our analysis they correspond to the pairwise hyperbolic distances between the nodes in the hyperbolic disk, which are provided in output by the nPSO model[9,10] when generating a network (see "Methods" section related to the nPSO model).

In the case of GSP, they correspond to the weighted shortest paths computed using the Johnson's algorithm[16] on the weighted network, where the connections are weighted by distances between the node pairs. If known, such distances can correspond to the geodesics, otherwise the weights can also represent other types of distances. The length of the GSP is equal to the sum of the distances of the connections involved in such a path.

The computation of the $\overline{\mathrm{pTSP}}(i, j)$ requires to find all the possible TSP between $(i, j)$. First of all, we apply the Johnson's algorithm[16] on the unweighted network to obtain the length of the TSP for all the node pairs (this is a one-time computation, not needed for each pair individually). Then, we apply the algorithm to compute the $\overline{\mathrm{pTSP}}$ between all node pairs, which is presented as pseudocode in Supplementary Note 2 and discussed in a dedicated "Methods" section.

A basic requirement of previous measures based on navigability is to have a connected network (or to consider the largest connected component), since it is meaningful to measure the navigability only between connected parts. The same basic requirement applies also to the measure of geometrical congruence. This guarantees that RD$(i, j)$ and $\overline{\mathrm{pTSP}}(i, j)$ are finite quantities.

### Greedy routing efficiency (GRE)
Given a network with $n$ nodes and $e$ edges, the greedy routing efficiency (GRE)[17,18] with respect to the set $\widetilde{E}$ of nonadjacent node pairs $(i, j)$ is

$$\mathrm{GRE}(\mathrm{pGRP}, \mathrm{RD}) = \left(\frac{1}{n \cdot (n-1) - 2 \cdot e}\right) \cdot \sum \frac{\mathrm{RD}(i,j)}{\mathrm{pGRP}(i,j)}; \text{with} (i,j) \in \widetilde{E}$$

In practice, for each pair $(i, j)$ of nonadjacent nodes we compute the ratio between RD$(i, j)$, which is a reference distance, and pGRP$(i, j)$, which is the projection of the greedy routing path between $(i, j)$ and set to infinite when the greedy routing is unsuccessful. The GRE is obtained as the average of such ratios, assuming values between 0 and 1. Note that pGRP$(i, j)$ can be different (asymmetric) from pGRP$(j,i)$, therefore we evaluate both pairs $(i, j)$ and $(j,i)$, resulting in a total of $n \cdot (n-1) - 2 \cdot e$ directed node pairs to be averaged. RD$(i, j)$ can be any node pairwise reference distance. In this study, in one case we consider RD(i,j) equal to the geodesic (GEO) and in another case equal to the geometrical shortest path (GSP), for more details please refer to the previous section on geometrical congruence (GC).

In the measure of greedy routing efficiency introduced in the original publications[17,18,21], all the $n \cdot (n-1)$ node pairs have been considered, both adjacent and nonadjacent nodes, and the associated formula is

$$\mathrm{GRE}(\mathrm{pGRP}, \mathrm{RD}) = \left(\frac{1}{n \cdot (n-1)}\right) \cdot \sum \frac{\mathrm{RD}(i,j)}{\mathrm{pGRP}(i,j)}; \text{with} (i,j) \in E$$

where $E$ is the set of pairs $(i, j)$ of both adjacent and nonadjacent nodes. We note that in the GRE formulation adopted in this study only nonadjacent node pairs have been considered, in order to guarantee a fair comparison of the values with respect to the measure of geometrical congruence (GC), which is also evaluated only for nonadjacent node pairs. In addition, we clarify that in the previous publications[17,18], in which we originally introduce GRE, the formula was given only for the special case that pGRP is unweighted (hence it is the number of the greedy routing hops between two nodes) and RD = TSP. Here instead

we propose a more general formula based on any reference distance that, according to the context of the scientific study, can be adequately selected.

## Computation of the $\overline{\mathrm{pTSP}}$ between all node pairs: algorithm design

We recall that the $\overline{\mathrm{pTSP}}$ is the average projection of the topological shortest paths between two nodes. Therefore, in order to compute the $\overline{\mathrm{pTSP}}$ between all node pairs, we need to find all the topological shortest paths between all node pairs.

Let us analyze a first solution computing the $\overline{\mathrm{pTSP}}$ for each individual node pair $(i, j)$. This solution can be implemented with a recursive approach. Starting from source node $i$ and performing a recursive visit to the neighbors (while avoiding loops) up to target node $j$, it is guaranteed to traverse once and only once all the topological shortest paths from $i$ to $j$. The projection (geometrical length) of such paths can be stored while traversing them, in order to compute their average, i.e. the $\overline{\mathrm{pTSP}}(i, j)$, at the end of the recursion process. If this recursion procedure is repeated for each node pair $(i, j)$, the result is the $\overline{\mathrm{pTSP}}$ between all node pairs. This *naïve* solution contains plenty of redundant computation that could be avoided with a smarter approach, which is still brute-force (in the sense that all node pairs are considered) and is analyzed below.

The brute-force version is based on introducing the concept of maximum depth. Let us consider the computation of $\overline{\mathrm{pTSP}}(i, j^*)$, where $j^*$ is a node at the maximum topological shortest paths from $i$: $\mathrm{TSP}(i, j^*) = \max_{j \in [1,N], j \neq i} \mathrm{TSP}(i, j)$. Starting from source node $i$ the neighbors are recursively visited until reaching the target node $j^*$ at recursion depth equal to $\mathrm{TSP}(i, j^*)$. Let us now consider the computation of $\overline{\mathrm{pTSP}}(i, k)$, where $k$ is any other node. Starting from source node $i$ the neighbors are recursively visited until reaching the target node $k$ at recursion depth equal to $\mathrm{TSP}(i, k) \leq \mathrm{TSP}(i, j^*)$. The key concept to understand is the following: all the paths visited in this recursion have been already visited during the previous computation of $\overline{\mathrm{pTSP}}(i, j^*)$, therefore we could have computed also $\overline{\mathrm{pTSP}}(i, k)$ while computing $\overline{\mathrm{pTSP}}(i, j^*)$. In other words, during the recursive procedure from source $i$ to a node $j^*$ at the maximum topological shortest paths from $i$, not only it is guaranteed to traverse once and only once all the topological shortest paths from $i$ to $j^*$, but it is also guaranteed to traverse once and only once all the topological shortest paths from $i$ to all the other nodes $k$. This has a significant implication to improve the naïve solution for computing the $\overline{\mathrm{pTSP}}$ between all node pairs: instead of performing one recursive procedure for each node pair $(i, j)$ with depth $\mathrm{TSP}(i, j)$ and compute $\overline{\mathrm{pTSP}}(i, j)$, it is sufficient to perform one recursive procedure with maximum depth $\mathrm{TSP}(i, j^*)$ for each node $i$ and compute $\overline{\mathrm{pTSP}}(i, j)$ for all $j \in [1, N], j \neq i$ at once. In few words, the brute-force algorithm avoids redundantly compute any $\mathrm{TSP}(i, k) \leq \mathrm{TSP}(i, j^*)$, whereas the naïve algorithm redundantly compute $\mathrm{TSP}(i, k)$ in a explicit way as a node pair and in an implicit way as an intermediate node pair on other $\mathrm{TSP}(i, j) \geq \mathrm{TSP}(i, k)$. We refer to this algorithmic solution as *brute-force* variant, since for each source node the recursion depth is always set to its maximum shortest path, implying that the $\mathrm{TSP}(i, j)$ for all node pairs are computed without considering a further possible optimization, which we discuss next.

Let us consider the *brute-force* variant and an arbitrary order of the source nodes $n_1, n_2, \ldots, n_N$. The first source node $n_1$ has recursion depth equal to $L_{n_1} = \max_{j \in [1,N], j \neq 1} \mathrm{TSP}(n_1, n_j)$ and it computes all $\overline{\mathrm{pTSP}}(n_1, n_j)$ with $j \in [1, N], j \neq 1$. The second source node $n_2$ has recursion depth equal to $L_{n_2} = \max_{j \in [1,N], j \neq 2} \mathrm{TSP}(n_2, n_j)$ and it computes all $\overline{\mathrm{pTSP}}(n_2, n_j)$ with $j \in [1, N], j \neq 2$. However, because of the symmetric property $\overline{\mathrm{pTSP}}(i, j) \equiv \overline{\mathrm{pTSP}}(j, i)$, the second source node $n_2$ does not

actually need to compute $\overline{\mathrm{pTSP}}(n_2, n_1)$, because the first source node $n_1$ already computed $\overline{\mathrm{pTSP}}(n_1, n_2)$. This also implies that $n_2$ does not need to consider $n_1$ in the computation of the recursion depth, and can set it instead as $L_{n_2} = \max_{j=3,\ldots,N} \mathrm{TSP}(n_2, n_j)$. This optimization in which the $\overline{\mathrm{pTSP}}$ for the symmetric pair is not computed again can be interpreted as *memoization*: an optimization technique ensuring that a function does not run for the same inputs more than once, by keeping a record of the results for the given inputs. In general, in the *brute-force* variant, the $i$th source node $n_i$ has recursion depth equal to $L_{n_i} = \max_{j \in [1,N], j \neq i} \mathrm{TSP}(n_i, n_j)$ and it computes all $\overline{\mathrm{pTSP}}(n_i, n_j)$ with $j \in [1, N], j \neq i$. Instead, in the *memoization* variant, the $i$th source node $n_i$ has recursion depth equal to $L_{n_i} = \max_{j=i+1,\ldots,N} \mathrm{TSP}(n_i, n_j)$ and it computes all $\overline{\mathrm{pTSP}}(n_i, n_j)$ with $j = i + 1, \ldots, N$. The computational advantage of the *memoization* variant comes from the following: since every subsequent source node does not consider the previous ones in the computation of the maximum topological shortest path, the recursion depth will be either equal or lower with respect to the brute-force variant. When it is lower, redundant $\mathrm{TSP}(n_i, n_j)$ computation is avoided because the recursion depth is adapted to the arbitrary order of the source nodes $n_1, n_2, \ldots, n_N$. We clarify that the *memoization*, intended as the fact that the $\overline{\mathrm{pTSP}}$ for symmetric pairs is only considered once, is present regardless of whether the recursion depth will be equal or lower, and it represents the first contribution in our algorithmic solution. Note that the recursions starting from different source nodes are independent, therefore they can be computed in parallel. This is possible because the maximum recursion depth for each source node is assigned before the recursion starts.

Now we take a step further in the analysis and discuss the choice of the arbitrary order of the source nodes. On one side, any permutation would provide the same results of the $\overline{\mathrm{pTSP}}$ between all node pairs. On the other side, different permutations can imply different recursion depths to be computed for the source nodes, therefore different computational times. In particular, the more the first source nodes have many topological shortest paths of high length, the higher the likelihood that following nodes will have a decreased recursion depth, since those long paths will be removed from the computation of the maximum. Given this, we need to find a criterion to select a permutation of nodes that would lead to a computational time advantage.

Theoretically, one could define an objective function (representing the computational load) as function of the recursion depths of the source nodes, then find the exact sequence of source nodes that minimizes it. However, considering that there are $N!$ permutations of the nodes to test, this solution is unfeasible in practice. For this reason, we decided to adopt a heuristic approach that provides an effective solution and consists in ordering the source nodes by decreasing average topological shortest path, which can be computed efficiently and guarantees that the nodes with many long paths are positioned first. This is the second contribution in our algorithmic solution, which we call *prioritization*. We highlight that choosing an optimized order of the source nodes (prioritization) is strictly synergetic with the memoization, whereas it would not have any effect on the brute-force approach.

In order to keep independent the recursions of different source nodes, which allows to process them in parallel, the prioritization step and the computation of the recursion depths for all source nodes can be performed in advance before any recursion. This pre-processing, in which we set the recursion depths and potentially reduce them with respect to the brute-force variant, results in a pre-pruning of the brute-force recursion tree.

At last, we would like to discuss potential questions that might arise. In particular, we can notice that while performing the recursion starting from a certain source node $i$ up to reaching its recursion depth, some intermediate paths could be topological shortest paths

starting from a different node $j$. Here a question that might arise is why such paths are not stored in order to not be visited again when the recursion from source node $j$ will be performed. Unfortunately, there are multiple issues related to this problem. First of all, the space complexity. Storing for a later usage all the topological shortest paths that are visited during a recursion would require a significant amount of space, which would make the computation unfeasible for networks of large size. Second, storing information from the recursion of node $i$ and using it for the recursion of node $j$ introduces a dependency between recursions of different source nodes, implying that they cannot be processed in parallel. Third, it is not clear whether there is an algorithmic solution to indicate to the recursion from node $j$ to simply skip exactly those paths stored during the recursion from node $i$. Even if such algorithmic solution exists and the memory does not represent a problem, it is not guaranteed that the time to store all the intermediate topological shortest paths, plus the time to indicate to the next recursions to skip them, plus the fact that the computation is not parallelized, still would provide a running time advantage with respect to simply traversing those paths again.

### Nonuniform popularity-similarity optimization model (nPSO) and classical popularity-similarity optimization model (PSO)

The Popularity-Similarity-Optimization (PSO) model[11] is a recently introduced generative model for networks that is based on growing soft random geometric graphs in the hyperbolic space. In this model the networks evolve optimizing a trade-off between node popularity (abstracted by the radial coordinate) and similarity (represented by the angular distance). The PSO model can reproduce many structural properties of real networks: clustering, small-worldness (concurrent low characteristic path length and high clustering), node degree heterogeneity with power-law degree distribution and rich-clubness. However, being the nodes uniformly distributed over the angular coordinate, the model lacks a non-trivial community structure.

The nonuniform PSO (nPSO) model[9,10] is a recently introduced generative model for realistic networks that is based on growing soft random geometric graphs with tailored community organization in the hyperbolic space. It is a generalization of the PSO model that exploits a nonuniform distribution of nodes over the angular coordinate in order to generate networks characterized by communities, with the possibility to tune their number, size and mixing property. In this study, we adopted a Gaussian mixture distribution of angular coordinates, with communities that emerge in correspondence of the different Gaussians, and the parameter setting suggested in the original studies[9,10]. Given the number of components $C$, they have means equidistantly arranged over the angular space, $\mu_i = \frac{2\pi}{C} \cdot (i-1)$, the same standard deviation fixed to 1/6 of the distance between two adjacent means, $\sigma_i = \frac{1}{6} \cdot \frac{2\pi}{C}$, and equal mixing proportions, $\rho_i = \frac{1}{C} (i = 1 \dots C)$. The community memberships are assigned considering for each node the component whose mean is the closest in the angular space. The other parameters of the model are: $N$, the number of nodes; $m$, around half of the average node degree; $T$, the network temperature, inversely related to the clustering; $\gamma$, the exponent of the power-law degree distribution. Given the input parameters ($N$, $m$, $T$, $\gamma$, $C$), the nPSO model provides in output: the adjacency matrix of the network; the geometrical coordinates of the nodes in the hyperbolic disk, the community memberships of the nodes; the pairwise hyperbolic distances (geodesics) between the nodes. For details on the generative procedure please refer to the original studies[9,10]. The MATLAB code is publicly available at the GitHub repository: https://github.com/biomedical-cybernetics/nPSO_model.

### Structural connectomes data

The dataset includes tractography-based connectivity matrices of 614 healthy individuals (Male = 230, Female = 384) generated by the enhanced Nathan Kline Institute-Rockland Sample (NKI-RS; fcon_1000.projects.nitrc.org/indi/enhanced/)[30]. Streamline count adjacency matrices were constructed by counting the NOS that terminated in each region of interest of the Yeo network functional parcellation (114 cortical nodes)[31]. The whole dataset (n = 614) was used to assess sex differences at the brain network level. In addition, from this dataset we also extracted a subset of n = 438 connectivity matrices of individuals in two different age ranges: [7, 30] years old (n = 223) and [55,85] years old ($n$ = 215). Further details on this dataset are available in[23], which is the study that processed the connectomes and made them publicly available.

The NOS weights represent connection strengths, however our computation requires weights to represent distances between the nodes. Longer white matter projections are more expensive in terms of their material and energy costs, thus making brain regions that are spatially close more likely to be connected[32]. Following this rationale, two brain regions connected by a higher number of streamlines tend to be at lower distance. Therefore, for every edge (i,j), the weight has been reversed according to the following formula:

$$w^*(i,j) = \frac{1}{1 + w(i,j)}$$

where $w(i,j)$ is the original weight (NOS) between the adjacent nodes $i$ and $j$ and $w^*(i,j)$ represent the reversed weight which we consider for our brain connectomic analysis.

### Hardware and software

MATLAB code has been used for all the simulations. The computation was executed exploiting several server nodes of the High-Performance Computing (HPC) cluster of ZIH, TU Dresden, each with 512 GB RAM, 2x AMD EPYC CPU 7702 @ 2.0 GHz (2×64 cores).

### Reporting summary

Further information on research design is available in the Nature Portfolio Reporting Summary linked to this article.

### Data availability

The artificial network datasets generated and analyzed in this study can be reproduced using the original code of the nPSO model available here: https://github.com/biomedical-cybernetics/nPSO_model. The parameters to generate the networks are all disclosed in the study.

The macroscale structural MRI brain connectomes dataset is available directly from the original study of Faskowitz et al.[23]

### Code availability

The MATLAB code to compute GC and GRE measures, to reproduce the results of some main figures of the study, and to plot the networks native disk representation in the hyperbolic space, is publicly available at the GitHub repository: https://github.com/biomedical-cybernetics/geometrical_congruence.

The DOI of the current first release of the code is https://doi.org/10.5281/zenodo.7221662.

If available in the future, the updated versions of the code (i.e. the most recent release) will be associated with this link that always point to the latest https://zenodo.org/badge/latestdoi/550208431.

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

## Acknowledgements

The authors thank the Centre for Information Services and High Performance Computing (ZIH) of the TU Dresden; the BIOTEC System Administrators for their IT support; Gloria Marchesi, Claudia Matthes and Sabine Zeissig for the administrative assistance at BIOTEC; YuanYuan Song, Zongzong Ji, Yining Xin, Weijie Guan and Lixia Huang for the administrative support at THBI; Hao Pang for the IT support at THBI. Work in the CVC's Biomedical Cybernetics Lab was supported by the independent research group leader running funding of the Technische Universität Dresden. Work in the CVC's Center for Complex Network Intelligence was supported by the Zhou Yahui Chair professorship of Tsinghua University, the starting funding of the Tsinghua Laboratory of Brain and Intelligence, and the National High-level Talent Program of the Ministry of Science and Technology of China.

## Author contributions

C.V.C. conceived the geometrical congruence and the content of the study. Both the authors contributed to design the algorithm, computational experiments, figures and items. AM implemented the code for the computation of GEO, GSP, pTSP and GRE; CVC prototyped the code for Figs. 2 and 3. A.M. finalized the computational analysis and realized figures and items. Both the authors analyzed and interpreted the results. C.V.C. wrote the main section of the article and section 1 of the Supplementary Information, and A.M. corrected them. A.M. wrote the "Methods" section and the other Supplementary Information sections, and C.V.C. corrected them. C.V.C. planned, directed and supervised the study.

## Competing interests

The authors declare no competing interests.
