## [Peer Review File · Nature Communications]

Geometrical congruence, greedy navigability and myopic transfer in complex networks and brain connectomesReviewers' Comments:

Reviewer #1:

Remarks to the Author:

This manuscript analyzes the efficiency of navigation in hyperbolic networks. The authors propose a new measure of geometrical congruence between topologically shortest paths (TSP) and either geodesics or geometrically shortest paths (GSPs). The authors find that TSPs deviate strongly from geodesics but are similar to the GSPs. Then, the authors use comparison between TSPs and GSPs to characterize differences in brain connectivity based on gender and age, finding larger deviations in females and in the aged group.

There are a number of problems with the analysis. First, the analyses are studied only for a given network size of 100 nodes. The previous results that the authors question are developed in the asymptotic regime of large networks.

Second, it is puzzling that the comparison when using the geodesics and the geometrically shortest paths results in such drastically different results, because these two types of the trajectories are very similar to each other, at least in the example in Figure 1b. This example in Figure 1b is used to motivate the study showing large differences between TSPs and GSPs/geodesics. The authors report that the differences between TSPs and GSPs are substantial but are not statistically significant between TSPs and geodesics. This is puzzling because GSPs and geodesics closely approximate each other (cf. Figure 1b).

Third, while the main criticisms of previous work made in this manuscript are based on the comparison between TSPs and geodesics, the authors use the comparison between TSPs and GSPs in their analysis of brain connectivity data. If that is the right measure, then this would support the previous work.

Fourth, the data analysis is not matched across groups. For example, there are almost twice as many data points for females than for males. This could be the reason for the observed differences between groups.

Fifth, the proposed measure of congruence is based on the ratio and so it could be strongly affected by the presence of infinite paths in either TSP and GSP.

Reviewer #2:

Remarks to the Author:

I read with great interest the paper by Carlo Vittorio Cannistraci and Alessandro Muscoloni on Geometrical congruence and efficient greedy navigability of complex networks

The paper is a pertinent contribution into the recently developing field of Network Geometry and I am in favour of publication of it with some minor amendments that in my view could help the readability of the paper

My main concern in this field is on the use of the "congruence" term. In ordinary geometrical sense this means that two objects (i.e. triangles) can be put one over the other, possibly after a mirroring. In this respect two geometric objects are congruent or not. The Geometric Congruence defined in this paper is instead a non-binary quantity determining the distance towards the required congruence. I believe that this does not help the reader to get the importance of the proposed quantity. I therefore urge the authors to find an alternative way to indicate their measure. (something like "hyperbolic similarity"? could it work?)

Given the topic, that builds on a recent preprint (Ref. 1) i suggest the author to spend few sentences in explaining to the large audience of the journal the definition of hyperbolic network and the papers dealing with this feature.

Why the peak for topological shortest path of length 4 are present only in Fig.2a?

I assume the error on the points figure 4 is much smaller than the separation of the two curves. Am I right?

Finally, I would start the paper straight with the congruence definition and its applications, I definitely would not start a paper with a "bad news", rather I would show how the new quantity helps in solving problems in Hyperbolic networks (obviously showing why it is necessary).

Point by point reply to Reviewers

Dear Editor and Reviewers, to simplify your review, all the modifications generated to address your comments are reported in red characters in the new revised text of the draft. You will find below the point by point reply (in blue color) to your comments (in black color). Whenever it was relevant, we reported portions of the text or figures within the replies (using a different font style). We fervidly thank you for your valuable support to improve the quality of our study.

Reviewer 1

This manuscript analyzes the efficiency of navigation in hyperbolic networks. The authors propose a new measure of geometrical congruence between topologically shortest paths (TSP) and either geodesics or geometrically shortest paths (GSPs). The authors find that TSPs deviate strongly geodesics but are similar to the GSPs. Then, the authors use comparison between TSPs and GSPs to characterize differences in brain connectivity based on gender and age, finding larger deviations in females and in the aged group.

There are a number of problems with the analysis. First, the analyses are studied only for a given network size of 100 nodes. The previous results that the authors question are developed in the asymptotic regime of large networks.

REPLY #1: We want to thank the Reviewer because, regardless of any misreading of the text provided in our study (we considered networks till 1000 nodes and not 100 as the Reviewer states), this comment raises an important and computational challenging question: what is happening when the node size N of the network grows and takes values larger than 1000 nodes?

Our reply is structured in two subpoints.

#1.1: We are sorry if we have to point out that the Reviewer comment is misleading when she/he says: << the analyses are studied only for a given network size of 100 nodes. >>.

It is true that the results in Fig. 1 and 2 - which we report as example to facilitate visualization - are for 100 nodes, but then in the next Figures of the study, we provide extended simulations with networks ranging the following parameters: community number $C = [0, 4]$; temperature $T = [0.1, 0.5, 0.7]$; average degree $\bar{d} = [4, 8, 12, 16, 20]$; power-law exponent $\gamma = [2, 2.25, 2.5, 2.75, 3]$; network node size $N = [100, \mathbf{1000}]$; 10 network realization for each parameter combination. This means a total of $2 \times 3 \times 5 \times 5 \times 2 \times 10 = 3000$ synthetic networks analyzed with two different network measures GC and GRE. **The figures with results related to $N = 1000$ are indeed in Suppl. Inf.**, but we commented them in the Results section (pages 5 and 6) of our previous manuscript: << **Fig. 3a-b results are confirmed also for networks with $N=1000$ and no fixed community organization (see Suppl. Fig. 1-4).** Hence, the second important finding of this study is that in general for $\gamma =]2,3]$ the hyperbolic networks generated with the nPSO model present medium to low congruence and greedy navigability, and that high congruence/navigability emerges only for γ proximal to 2 (at

least in these nPSO hyperbolic networks). >> Hence, it seems that the Reviewer missed or erroneously reported this important result of our study.

#1.2: Nevertheless, we interpreted the comment of the Reviewer as a spur to investigate and confirm the results of our study also for larger values of node size N in the networks, and we revised the study accordingly. Please note: we want to stress that this costed more than one year of work both to think/redesign an optimized and faster version of the algorithm for congruency and for running all the simulations. The results of this ‘tremendous’ effort, to move forward the frontiers of knowledge in network geometry, are now presented in the following new sections of the revised manuscript:

Results: “Geometrical congruence in networks with patent geometry: the case of hyperbolic networks” (Page 12)

<< [...] A possible objection to our result could be that the geometrical congruence might emerge in the asymptotic regime for growing node size N of the network and that in this study we offer evidence only till $N=1000$. However, addressing this concern is nontrivial because the computational complexity of the algorithm is onerously growing with N and γ . Therefore, to investigate this concern we had to develop an optimized and parallelised algorithm that is discussed in the next section. After months of computing we obtained results to investigate how the trend of $GC(\overline{pTSP}, \text{GEO})$ (Fig. 4a) and $GRE(pGRP, \text{GEO})$ (Fig. 4b) varies for growing N (till 10^5 with computation and 10^6 with extrapolation, see Fig. 4 legend for details) and $\gamma = [2, 2.25, 2.5, 2.75, 3]$. Since the computation is burdensome in time, the other parameters that are less important for our investigation were fixed to reference values: $\bar{d} = 12$, $T = 0.1$, and $C = 4$. More parameter combinations for networks of $N=10^4$ are provided in Suppl. Fig. 6-7. The results of this computation confirm that the trend of $GC(\overline{pTSP}, \text{GEO})$ (Fig. 4a) and $GRE(pGRP, \text{GEO})$ (Fig. 4b) is not growing with node size N . Therefore, the results we present discourage the notion (included in a review study on network geometry¹) that in the asymptotic regime the greedy navigation in hyperbolic networks with $\gamma < 3$ can always find ultrashort paths which are (hardly) congruent with the geodesics. This is the third important finding of our study. [...] >>

Results: “An algorithm to compute geometrical congruence in networks” (Page 13)

<< The essence of this section is on how to design an optimized algorithm that reduces 26 years of worst scenario computation to one week, and this represents the fourth important achievement of our study. Indeed, the \overline{pTSP} is the average projection of the topological shortest paths between two nonadjacent nodes in the network. Therefore, in order to compute the \overline{pTSP} , we need to find all the topological shortest paths for each nonadjacent node pair in the network that is a nontrivial computational task dependent on different topological features. [...] >>

Here we report only the incipit. This section is extensive and enriched with an elaborated time complexity analysis which is supported by “Figure 5. Running time estimation”. For this reason, we refer directly to check the entire section at Page 13 of revised manuscript.

Methods: “Computation of the \overline{pTSP} between all node pairs: algorithm design” (Page 25)

Please, refer directly to the entire section at Page 25 of revised manuscript.

Supplementary information: “Space and time complexity”

Please, refer directly to the entire section in Suppl. Inf.

Supplementary information: “Running time estimation”

Please, refer directly to the entire section in Suppl. Inf.

Supplementary information: “Suppl. Algorithm 1. Pseudocode to compute the \overline{pTSP} between all node pairs”

Please, refer directly to the entire section in Suppl. Inf.

Second, it is puzzling that the comparison when using the geodesics and the geometrically shortest paths results in such drastically different results, because these two types of the trajectories are very similar to each other, at least in the example in Figure 1b.

This example in Figure 1b is used to motivate the study showing large differences between TSPs and GSPs/geodesics.

REPLY #2: We really thank the Reviewer for this appropriate comment that helped us to significantly improve the message of the study. Yes, the Reviewer is right that the Figure 1b of the previous manuscript was not appropriate, and even misleading, to offer an example of the general case. Indeed, it was a rare degenerative case that occurs when the pairs of nodes are almost on the radius. Now in the new Fig. 1 we offer different examples and a discussion of the general and rare cases. In brief, in the new Fig. 1 we show that in the majority of cases (mode and mean of the distribution) the geodesic follows a path with large differences from the pTSPs and GSPs. This is in agreement with the large-scale simulations we report after in the article in Fig. 4, where for growing N we show large differences between pTSPs and geodesics only, and we offer evidence for high congruence between pTSPs and GSPs.

Figure 1. Visualization of geodesics, geometrical and topological shortest paths in hyperbolic space.

A nPSO network has been generated with parameters $N = 100$, $\bar{d} = 8$, $T = 0.1$, $\gamma = 2.5$ and $C = 5$. **(a)** Representation of the nPSO network in the hyperbolic disk: the links are in grey colour and follow the hyperbolic geodesics, the nodes are coloured by community membership and their size is proportional to the logarithm of the degree. **(b)** The histogram indicates, at different intervals, the proportion of GEO/\overline{pTSP} values for nonadjacent node pairs in the nPSO network. **(c)** The panel highlights the hyperbolic geodesic (GEO, red), the geometrical shortest path (GSP, green) and the links involved in all the possible topological shortest paths (TSP, black dashed) between two specific nonadjacent nodes in the network, characterized by a ratio $GEO/\overline{pTSP} = 0.61$ corresponding to the mode value. **(d)** Analogous to panel (c) but for two nonadjacent nodes characterized by a ratio $GEO/\overline{pTSP} = 0.67$ corresponding to the mean value. **(e)** Analogous to panel (c) but for two nonadjacent nodes characterized by a ratio $GEO/\overline{pTSP} = 0.96$ corresponding to a value in the top 5% of the histogram.

Figure 4. GC and GRE evaluation for increasing network size.

nPSO networks have been generated with parameters $N = [100, 1000, 10000, 100000]$, $\bar{d} = 12$, $T = 0.1$, $\gamma = [2, 2.25, 2.5, 2.75, 3]$ and $C = 4$. For each combination of parameters, 10 networks have been generated. For each network we have computed: **(a)** $GC(\overline{pTSP}, GEO)$, **(b)** $GRE(\rho GRP, GEO)$, **(c)** $GC(\overline{pTSP}, GSP)$, **(d)** $GRE(\rho GRP, GSP)$. Each plot reports 5 curves for the different values of γ , indicating the mean value (over 10 network realizations) of the respective network measure for increasing network size. Error bars are not reported since negligible. In order to provide an estimation for networks of size $N = 1000000$, for each curve we used the spline algorithm to perform interpolation at $N = [100, 1000, 10000, 100000]$ and extrapolation at $N = 1000000$, the resulting values are highlighted with dashed lines.

The authors report that the differences between TSPs and GSPs are substantial but are not statistically significant between TSPs and geodesics. This is puzzling between GSPs and geodesics closely approximate each other (cf. Figure 1b).

REPLY #3: We never reported that: << the differences between TSPs and GSPs are substantial >>. Actually we reported the opposite, see previous manuscript version Page 7: << the measures $GC(\overline{pTSP}, GSP)$ and $GRE(pGRP, GSP)$ show high congruence and high navigability for any parameter combination (Fig. 3c-d) at $T=0.1$. >>

Hence, here we cannot understand the point of the reviewer. As much as concern for the Fig. 1b, we already discussed in the REPLY #2 above that the example in Fig. 1 of the previous manuscript was a rare degenerative case that was not identifying the general trend, and that now in the new Fig. 1 we offer a careful discussion of the general and rare cases by plotting the distribution. These results are also confirmed in the new Fig. 4 for growing N , see REPLY #2 above.

Third, while the main criticisms of previous work made in this manuscript are based on the comparison between TSPs and geodesics, the authors use the comparison between TSPs and GSPs in their analysis of brain connectivity data. If that is the right measure, then this would support the previous work.

REPLY #4: The reply to this concern was already in the text of the previous manuscript. However, thanks to the Reviewer's concern, we revised adding few sentences that makes it even more explicit. The revised text is reported below with new text in red:

Results: "Geometrical congruence in networks with patent geometry: the case of hyperbolic networks" (Pages 12-13)

<< [...] Interestingly, if we change the reference distance from geodesic to GSP (the green line in Fig. 1b) then the measures $GC(\overline{pTSP}, GSP)$ and $GRE(pGRP, GSP)$ show high congruence and high navigability for any parameter combination (Fig. 3c-d) at $T=0.1$. **And this is confirmed in Fig. 4c and Fig. 4d also for growing N .** On one side, this result suggests that using the GSP as reference distance for GC and GRE is **misleading** if the goal is to **investigate** the geometrical congruence of a network with its underlying geometry **for different γ values**, in which case the geodesic should be used as reference. On the other side, the proposed GC measure seems properly designed, since it is strongly associated to GRE on hyperbolic networks with high clustering ($T=0.1$) and, as expected according to theory¹¹, when temperature increases ($T=0.3$ and $T=0.5$, see Suppl. Fig. 2-7), clustering decreases, hyperbolic geometry tend to vanish and $GC(\overline{pTSP}, GSP)$ has a tendency to deviate from $GRE(pGRP, GSP)$ which becomes more evident for $C=4$ and $N=1000$ (Suppl. Fig. 5). Therefore $GC(\overline{pTSP}, GSP)$ can still be considered an interesting marker to compare differences of geometrical congruence between structural and weighted connectivity across complex networks **with latent geometry such as brain connectomes as we will discuss below in the last section of the Results.** >>

In addition, to address the Reviewer's concern, we reorganized the manuscript separating in two different section the investigation of congruence in networks with patent and latent geometry. Please, refer to these two sections:

Results: “Geometrical congruence in networks with patent geometry: the case of hyperbolic networks”

Results: “Geometrical congruence in networks with latent geometry: the case of brain connectomes”

Fourth, the data analysis is not matched across groups. For example, there are almost twice as many data points for females than for males. This could be the reason for the observed differences between groups.

REPLY #5: We thanks the Reviewer for the useful concern. To address it, we added AUPR and trustworthiness p-value on the AUPR which are measures that correct for class unbalance in the data. The new results confirm the old ones. There are significant differences between groups also considering AUPR. See these sections below.

Results: “Geometrical congruence in networks with latent geometry: the case of brain connectomes” (Page 20)

<< [...] Results in Fig. 6a and Fig. 6d evidence that $GC(\overline{pTSP}, GSP)$ difference (considering NOS) is statistically significant (p-value<0.01, Mann-Whitney test) both for gender and age respectively. Although this result is significant, it does not say anything about the extent to which GC can be used as a marker that quantifies differences in brain connectomes and whether it works better than the greedy navigability efficiency - also known in network neuroscience as Efficiency Ratio (E_R) - proposed in previous studies^{22,23,32}. Therefore, we evaluated the extent to which GC and E_R can discriminate two states considering the area under precision recall curve (AUPR), which is robust against different group sizes³⁵. In addition, we evaluated whether a certain value of AUPR is statistically significant in comparison to a random permutation of the labels considering a measure called trustworthiness³⁶. [...] >>

Figure 6. GC as marker for gender and age discrimination in structural connectomes.

We have analysed a dataset of structural connectomes with gender and age annotation. For each network, we have computed the $GC(GSP, \overline{pTSP})$ and the E_R with connection weights based on NOS or 3D (see Methods for details). **(a)** We divided the connectomes in two groups related to male and female subjects, for each group we have estimated by kernel-density the probability density function of GC, which is reported in the subplot (male in black dashed line, female in red solid line): x-axis represents the $GC(GSP, \overline{pTSP})$ and y-axis the density function. The p-value of the Mann-Whitney test shows that GC can significantly discriminate between male and female connectomes (p-value < 0.01). **(b-c)** The barplots compare the AUPR of GC and E_R methods in discriminating between male and female connectomes, when using connection weights based on NOS (b) or 3D (c). An asterisk indicates that the AUPR-trustworthiness³⁶ with Bonferroni correction over the 4 markers is significant (p-value < 0.01). **(d)** We have repeated the same analysis as in (a), but considering two groups of connectomes related to two age-ranges of the subjects: [7-30] (black dashed line) and [55-85] (red solid line). The p-value of the Mann-Whitney test shows that GC can significantly discriminate between connectomes of young and elderly subjects (p-value < 0.01). **(e-f)** The barplots compare the AUPR of GC and E_R methods in discriminating between connectomes related to two age-ranges, when using connection weights based on NOS (e) or 3D (f). An asterisk indicates that the AUPR-trustworthiness with Bonferroni correction over the 4 markers is significant (p-value < 0.01).

In addition, all this section on brain connectomes was majorly expanded to reply to three questions.

Results: “Geometrical congruence in networks with latent geometry: the case of brain connectomes” (Page 19)

<< [...] In this study we focus our investigation on the comparison between greedy navigability and geometrical congruence in macroscale structural MRI brain connectomes, aiming to address three questions. The first is inherent to this network geometry study and regards whether we can provide any computational evidence that geometrical congruence, as theoretically expected, offers a concrete advantage on measuring properties of latent space networks with respect to greedy navigability. The second is an important open problem in network neuroscience³². Muscoloni et al.^{22,23} were the first to design a measure of network greedy navigability efficiency for complex networks in general, and Seguin et al.³² were the first to discuss and motivate its application on brain connectomes. Here we wish to make a step forward to investigate whether greedy navigability efficiency, and consequently also geometrical congruence, can be used as markers to differentiate different phenotypical states in brain connectomes. The third is a relevant debated problem at the interface between network geometry and network neuroscience. It regards whether navigability and congruence can be used to gain any evidence at support of the thesis that macroscale structural MRI brain connectomes might have a latent developmental neurobiological geometry which accounts for more information content than the 3D visible Euclidean geometry. Indeed, brain networks are associated to a developmental neurobiological geometry which also influenced by neuronal plasticity³³ and, according to some studies, might be approximated by hyperbolic distances²⁸⁻³⁰. However, as a matter of fact, it is latent because both the type of geometry and the coordinates of the nodes in this developmental neurobiological space are unknown. [...] >>

Fifth, the proposed measure of congruence is based on the ratio and so it could be strongly affected by the presence of infinite paths in either TSP and GSP.

REPLY #6: Respected Reviewer, we thank you for offering the opportunity to explicitly explain that previous measures of navigability, and consequently the congruence, are only applied to connected networks. Therefore infinite paths are not present. We clarified this in the main text and in the Methods section.

Results: “A mathematical expression to measure geometrical congruence in networks” (Page 11)

<< A basic requirement of previous measures based on navigability is to have a connected network (or to consider the largest connected component), since it is meaningful to measure the navigability only between connected parts. The same basic requirement applies also to the measure of geometrical congruence. >>

Reviewer 2

I read with great interest the paper by Carlo Vittorio Cannistraci and Alessandro Muscoloni on Geometrical congruence and efficient greedy navigability of complex networks

The paper is a pertinent contribution into the recently developing field of Network Geometry and I am in favour of publication of it with some minor amendments that in my view could help the readability of the paper

My main concern in this field is on the use of the “congruence” term. In ordinary geometrical sense this means that two object (i.e triangles) can be put one over the other, possibly after a mirroring. In this respect two geometric object are congruent or not. The Geometric Congruence defined in this paper is instead a non-binary quantity determining the distance towards the required congruence. I believe that this does not help the reader to get the importance of the proposed quantity. I therefore urge the authors to find an alternative way to indicate their measure. (something like “hyperbolic similarity”? could it work?)

REPLY #1: We thank the Reviewer for the positive comments. We took in serious consideration this main concern by largely revising the text of the manuscript as follow:

Results: “The geometrical congruence problem” (Page 6)

<< Up today, in network geometry, the notion of congruence was visually and qualitatively introduced when the projections of the topological shortest paths (pTSP) follow closely their associated geodesic in the underlying space^{1,11}. However, as we stressed in the introduction, a quantitative computational measure of geometrical congruence is missing in network science, and this study aims to fill this conceptual gap.

A first concern to address is on the use of the ‘congruence’ term. In mathematics, the ordinary geometrical sense of this word means that two objects, when they are superimposed (possibly after a mirroring operation), are perfectly aligned. In this respect two geometric objects are either congruent or not, and therefore the attribute associated to the term congruence in mathematics is a binary quantity: presence or absence. When this condition is satisfied, we will call it explicitly ‘hard’ congruence. Hard in the sense of binary condition because the etymological meaning of congruence does not imply this hard interpretation. Indeed, the term congruence comes from Latin ‘*congruens -entis*’ and it is believed to be composed of two words ‘cum’ (together) and ‘(g)ruere’ (to move/to come fast ; whereas the ‘g’ is a guttural sound that has the intention to reinforce). Therefore, congruence reads as “to come very fast together”, which can be used not necessarily in a binary manner but also in a relaxed form, in order to express in general the extent to which the trajectory of two paths run close together. [...] >>

In addition, please note that the definition of congruence that we introduce and investigate is not exclusively designed for hyperbolic networks, but it is valid for any patent or latent geometry. In order to make this even more clear, we performed the following actions.

We included this text in the Introduction section (Pages 5-6):

<< [...] Then, we proceed to measure geometrical congruence and greedy navigability efficiency in networks with patent (the contrary of latent in *Latin*) geometrical space, in the sense that the coordinates of the network nodes in this space are known. Specifically, we investigate hyperbolic networks generated with the nonuniform popularity-similarity optimization (nPSO) model^{13,14} which, as clarified above, is a generalization of the PSO model¹² able to grow realistic hyperbolic soft random geometric graphs with tailored community structure. Finally, we show how, contrary to the greedy navigability measures that need nodes geometrical coordinates (patent geometrical space, known or inferred by network embedding), the proposed geometrical congruence measure can be applied as a marker to reveal differences in real networks with latent geometrical space such as brain connectomes, without the requirement of nodes geometrical coordinates, but only geometrical weights (presumably associated with the latent space). Indeed, brain connectomes topological weights are associated to a developmental neurobiological geometry which, according to some studies, might be approximated by hyperbolic distances²⁸⁻³⁰. However, as a matter of fact, the geometry is latent because both the type of geometry and the coordinates of the nodes in this developmental neurobiological space are unknown. >>

We reorganized the manuscript separating in two different sections the investigation of congruence in networks with patent and latent geometry. Please, refer to these two sections:

Results: “Geometrical congruence in networks with patent geometry: the case of hyperbolic networks”

Results: “Geometrical congruence in networks with latent geometry: the case of brain connectomes”

Given the topic, that builds on a recent preprint (Ref. 1) I suggest the author to spend few sentences in explaining to the large audience of the journal the definition of hyperbolic network and the papers dealing with this feature.

REPLY #2: Again, we thank the Reviewer for this useful suggestion that we took in seriously consideration taking the inspiration to draw an extended story line that can help many to follow how the field evolved from the origin of random geometrical graph to current network geometry and hyperbolic network. Therefore, we included this text below in the Introduction section (Pages 2-4).

<< [...] If the shortest paths provide the ‘syntax’ of the interactions between two points, their projections in the respective underlying geometrical space reveal the ‘semantics’: the meaning behind those interactions. Not surprisingly, studies in network geometry¹ suggest that the latent geometrical space behind the observable topology of a network drives the navigation² of the network structure according to the distances between nodes in the geometrical space³. This means that the process behind the formation of connectivity in many networked complex systems follows a rule of association between the latent variables of the system that can be described by a manifold in a geometrical space. This geometrical space determines also the type of network. In relation with the global curvature of the manifold we can have three different types of geometries: hyperbolic (negative curvature), Euclidean (zero curvature), elliptic (positive curvature). Likewise, in network geometry we can encounter: hyperbolic networks, Euclidean networks and elliptic networks. Different geometries can trigger different structural and functional features of the associated networks.

Network geometry is currently rising as a compelling research area in physics¹, however we should mention that it received a considerable legacy of concepts and analytical methods from random geometrical graph theory. In 1961, Gilbert proposed a model⁴ of random geometric graphs according to which nodes (vertices) are placed following a Poisson point process, and links (edges) are formed between those within a fixed range. Indeed, in graph theory, a hard random geometric graph (hRGG) is a spatial network⁵: an undirected graph constructed by randomly placing N nodes in some metric space (according to a specified probability distribution) and connecting two nodes by a link if and only if their distance is within a given range, e.g. smaller than a certain neighbourhood radius. The adjective ‘hard’ means that this threshold on the distance is deterministic and therefore hard. Random geometric graphs resemble real networks in a number of ways. For instance, they spontaneously demonstrate community structure (clusters of nodes with high modularity)⁶. Other types of non-geometric random graphs, such as those generated using the Erdős–Rényi (ER)⁷ model or Barabási–Albert (BA)⁸ model, do not spontaneously create this structural feature. Indeed, in absence of geometry, it requires a more complicated modelling strategy such as the stochastic block model⁹, where the community structure is not spontaneously emerging but is induced by design.

In 1988 Waxman¹⁰ generalised the standard RGG in the Euclidean space by introducing a probabilistic connection function as opposed to the deterministic one suggested by Gilbert. This type of RGG with probabilistic connection function is often referred to as soft random geometric graph (sRGG), which has two sources of randomness: the location of nodes and the formation of links. Indeed, the adjective ‘soft’ indicates that the threshold on the distance is probabilistic and therefore soft. Waxman introduced the soft threshold to control the node clustering, a structural connectivity feature of the RGG, and to use this network as benchmark to study how some functional properties of wireless networks change according to the topology. Indeed, Waxman’s paper addresses the problem of routing connections in a large scale packet switched network supporting multipoint communications. All these studies focused on networks in the Euclidean space using a node-static model: the model evolves

growing links between a static ensemble of geometrically located nodes. We had to wait till 2010 for the first non-Euclidean model, when Krioukov et al.¹¹ proposed a static model for hyperbolic networks and discussed its relevance for information transport by greedy navigation, such as in the Internet. Important modelling advancements occurred in 2012 when Papadopoulos et al.¹² introduced the Popularity-Similarity optimization (PSO) model of hyperbolic networks, which is the first growing model: the model evolves geometrically adding a new node at the periphery of the network and its links probabilistically attach to the rest of the nodes according to the geometrical distance. In 2018 Muscoloni and Cannistraci introduced a generalization of the PSO termed nonuniform PSO (nPSO)¹³. The nPSO allows to design a tailored network community structure in the hyperbolic disk and can be used as benchmark for testing algorithms for community detection^{14–16}, link prediction^{14,17,18}, network embedding^{19,20}, hyperedge entanglement²¹, or for evaluating the extent to which the efficiency of greedy navigability²² is impacted by the topological modifications controlled by means of the underlying hyperbolic geometry²³. >>

Why the peak for topological shortest path of length 4 are present only in Fig.2a?

REPLY #3: We thank the Reviewer for this concern because it is indeed a key point about sparsity. This happens because that is the only subplot with parameter $d=4$, so the network is very sparse and therefore the path lengths are longer. We added the following text in the revised manuscript and a SI figure to explain this better.

Results: “The geometrical congruence problem” (Page 9)

<< The peak for topological shortest paths of length 4 are present only in Fig. 2a because that is the only subplot with parameter $d=4$, the network is very sparse, and therefore the path lengths are longer. In Suppl. Fig. 1 we provide further evidence about this phenomenon induced by sparsity. >>

Suppl. Figure 1. TSP length distribution in hyperbolic networks.

(a) The panel is equivalent to panel (a) of Fig. 2, showing the comparison of GEO and \overline{pTSP} distributions in nPSO networks with varied $\bar{d} = [4, 12, 20]$. (b) nPSO networks have been generated with parameters $N = 100$, $\bar{d} = [4, 12, 20]$, $T = [0.1, 0.5, 0.9]$, $\gamma = [2, 2.5, 3]$ and $C = 4$. For each network, we computed the TSP for all nonadjacent node pairs and then assessed the proportion of pairs characterized by each TSP length from 2 to 8 (the maximum TSP length among the networks generated). The results of such TSP proportions are reported in 3 heatmaps corresponding to $\bar{d} = [4, 12, 20]$, every row represents a different combination of parameters T and γ and every column a TSP length.

I assume the error on the points figure 4 is much smaller than the separation of the two curves. Am I right?

REPLY #4: We thank the Reviewer for the useful concern. To address it, we added AUPR and trustworthiness p-value on the AUPR which are measures that correct for class unbalance in the data. The new results confirm the old ones. There are significant differences between groups also considering AUPR. See these sections below.

Results: “Geometrical congruence in networks with latent geometry: the case of brain connectomes” (Page 20)

<< [...] Results in Fig. 6a and Fig. 6d evidence that $GC(\overline{pTSP}, GSP)$ difference (considering NOS) is statistically significant (p-value<0.01, Mann-Whitney test) both for gender and age respectively. Although this result is significant, it does not say anything about the extent to which GC can be used as a marker that quantifies differences in brain connectomes and whether it works better than the greedy navigability efficiency - also known in network neuroscience as Efficiency Ratio (E_R) - proposed in previous studies^{22,23,32}. Therefore, we

evaluated the extent to which GC and E_R can discriminate two states considering the area under precision recall curve (AUPR), which is robust against different group sizes³⁵. In addition, we evaluated whether a certain value of AUPR is statistically significant in comparison to a random permutation of the labels considering a measure called trustworthiness³⁶. [...] >>

Figure 6. GC as marker for gender and age discrimination in structural connectomes.

We have analysed a dataset of structural connectomes with gender and age annotation. For each network, we have computed the $GC(GSP, \overline{pTSP})$ and the E_R with connection weights based on NOS or 3D (see Methods for details). **(a)** We divided the connectomes in two groups related to male and female subjects, for each group we have estimated by kernel-density the probability density function of GC, which is reported in the subplot (male in black dashed line, female in red solid line): x-axis represents the $GC(GSP, \overline{pTSP})$ and y-axis the density function. The p-value of the Mann-Whitney test shows that GC can significantly discriminate between male and female connectomes (p-value < 0.01). **(b-c)** The barplots compare the AUPR of GC and E_R methods in discriminating between male and female connectomes, when using connection weights based on NOS (b) or 3D (c). An asterisk indicates that the AUPR-trustworthiness³⁶ with Bonferroni correction over the 4 markers is significant (p-value < 0.01). **(d)** We have repeated the same analysis as in (a), but considering two groups of connectomes related to two age-ranges of the subjects: [7-30] (black dashed line) and [55-85] (red solid line). The p-value of the Mann-Whitney test shows that GC can significantly discriminate between connectomes of young and elderly subjects (p-value < 0.01). **(e-f)** The barplots compare the AUPR of GC and E_R methods in discriminating between connectomes related to two age-ranges, when using connection weights based on NOS (e) or 3D (f). An asterisk indicates that the AUPR-trustworthiness with Bonferroni correction over the 4 markers is significant (p-value < 0.01).

In addition, all this section on brain connectomes was majorly expanded to reply to three questions.

Results: “Geometrical congruence in networks with latent geometry: the case of brain connectomes” (Page 19)

<< [...] In this study we focus our investigation on the comparison between greedy navigability and geometrical congruence in macroscale structural MRI brain connectomes, aiming to address three questions. The first is inherent to this network geometry study and regards whether we can provide any computational evidence that geometrical congruence, as theoretically expected, offers a concrete advantage on measuring properties of latent space networks with respect to greedy navigability. The second is an important open problem in network neuroscience³². Muscoloni et al.^{22,23} were the first to design a measure of network greedy navigability efficiency for complex networks in general, and Seguin et al.³² were the first to discuss and motivate its application on brain connectomes. Here we wish to make a step forward to investigate whether greedy navigability efficiency, and consequently also geometrical congruence, can be used as markers to differentiate different phenotypical states in brain connectomes. The third is a relevant debated problem at the interface between network geometry and network neuroscience. It regards whether navigability and congruence can be used to gain any evidence at support of the thesis that macroscale structural MRI brain connectomes might have a latent developmental neurobiological geometry which accounts for more information content than the 3D visible Euclidean geometry. Indeed, brain networks are associated to a developmental neurobiological geometry which also influenced by neuronal plasticity³³ and, according to some studies, might be approximated by hyperbolic distances^{28–30}. However, as a matter of fact, it is latent because both the type of geometry and the coordinates of the nodes in this developmental neurobiological space are unknown. [...] >>

Finally, I would start the paper straight with the congruence definition and its applications, I definitely would not start a paper with a “bad news”, rather I would show how the new quantity helps in solving problems in Hyperbolic networks (obviously showing why it is necessary).

REPLY #5: Yes, thanks, we understood the point of the Reviewer and indeed we reshaped the article according to this suggestion. We introduced these new sections in the revised manuscript:

Results: “The geometrical congruence problem”

Results: “A mathematical expression to measure geometrical congruence in networks”

Results: “Geometrical congruence in networks with patent geometry: the case of hyperbolic networks”

Reviewers' Comments:

Reviewer #1:

Remarks to the Author:

In the revised manuscript, the authors' have made a good effort to check the validity of their results for larger networks. Nevertheless, the manuscript remains misleading in the following two major ways.

1) One of the main results in the manuscript aims to disprove the results by Krioukov and collaborators that topologically shortest paths (TSP) are similar to geodesics in the underlying hyperbolic space. The authors show that TSP are not similar to geodesics but are similar to geometric shortest paths (GSP). However, it could be argued that GSP is the metric that Krioukov used. The algorithm there was to find a neighboring node with the shortest hyperbolic distance and iterate until the destination node is reached. This is the algorithm behind the GSP used in this paper. Therefore, the main result in this manuscript disputes not the finding by Krioukov but rather the misinterpretation of it. The geometrical congruence (GS) introduced here is for all practical purposes GSP. This removes one of the main novel contribution in the manuscript. The algorithm for computing shortest paths might be of interest, as is analysis of the brain connectome. However, the manuscript need to be shorten by a factor of two at least to focus the exposition to novel results.

2) In the analysis of brain connectome data, the authors show that the latent space is not a three-dimensional Euclidean space. An obvious question that is left unanswered is whether it is hyperbolic.

More minor comments follow:

The abstract states that "hyperbolic networks do not demonstrate high congruence" but does not specify congruence with what. If the authors find that hyperbolic networks are not congruent with themselves, then this indicates a problem with their analyses. In particular, the networks are generated here in a specific way based on the authors' prior published work.

The abstract uses the term "patent geometry" without ever defining it. How this is different from the standard term "latent geometry" is not described sufficiently well. There is a mention of this on line 143, arguing that this the contrary of latent in Latin. A mathematical definition of how patent differs from latent space would be helpful.

The introductory opening to the manuscript (line 38-47) is not relevant to the science described in the manuscript. It would be better to use a different opening that does not use the words "hatred" and "treachery".

Similarly, the term "memoization" introduced as part of the proposed algorithm has a very specific meaning in computer science. It would be better not to use it here. Memorization would be easier for readers and more appropriate

Line 514, instead of "two information" it might be better to state "two pieces of information" or "two properties".

Line 537: it is not appropriate to refer to the authors own work as "the first" given that these publications (refs. 22 and 23) are from 2017 and 2019 whereas Krioukov and collaborators have been publishing on this topic since at least 2008.

Reviewer #2:

Remarks to the Author:

I read the revised version of the paper that in my opinion has greatly improved. The authors answered with great detail to my questions. I am satisfied by their work. I ask publication of the paper in the present form

REPLY TO REVIEWER COMMENTS

Reviewer #1 (Remarks to the Author):

In the revised manuscript, the authors' have made a good effort to check the validity of their results for larger networks. Nevertheless, the manuscript remains misleading in the following two major ways.

1) One of the main results in the manuscript aims to disprove the results by Krioukov and collaborators that topologically shortest paths (TSP) are similar to geodesics in the underlying hyperbolic space. The authors show that TSP are not similar to geodesics but are similar to geometric shortest paths (GSP). However, it could be argued that GSP is the metric that Krioukov used. The algorithm there was to find a neighboring node with the shortest hyperbolic distance and iterate until the destination node is reached. This is the algorithm behind the GSP used in this paper. Therefore, the main result in this manuscript disputes not the finding by Krioukov but rather the misinterpretation of it.

REPLY: This comment is invalid because the Reviewer makes confusion between the greedy routing path (GRP) and the geometrical shortest path (GSP). The definition of GSP that the Reviewer provides is: << However, it could be argued that GSP is the metric that Krioukov used. The algorithm there was to find a neighboring node with the shortest hyperbolic distance and iterate until the destination node is reached. >> But the algorithm that the Reviewer describes is greedy and corresponds with GRP and not GSP. The GRP is indeed, and without any doubt, the metric that Krioukov et al. used to constrain their evaluations:

From: Hyperbolic geometry of complex networks. D Krioukov, F Papadopoulos, M Kitsak, A Vahdat, M Boguná Physical Review E 82 (3), 036106 (2010)

Page 82:

<< To estimate the GF [greedy forwarding] efficiency in static networks, we compute the following metrics: (i) the percentage of **successful paths** which is the proportion of paths that reach their destinations; (ii) the average hop length of **successful paths**; and (iii) the average and maximum stretch of **successful paths**. We consider three types of stretch. The first stretch is the standard hop stretch defined as the ratio between the hop lengths of greedy paths and the corresponding shortest paths in the graph. We denote its average and maximum by s_1 and $\max(s_1)$. The optimal paths have stretch equal to 1. The other two stretches are hyperbolic. They measure the deviation of the hyperbolic length, **traveled by a packet along either the greedy or shortest path**, from the hyperbolic distance between the source and destination. [...] For greedy paths, we denote the average and maximum of this stretch by s_2 and $\max(s_2)$; for shortest paths those are denoted by s_3 and $\max(s_3)$. >>

All the measure described here are based on the GRP except s3 and maxs3 that are based on GSP but are defined only and exclusively for the node pairs for which a successful GRP path exists. The Reviewer was induced in confusion by the fact that s2 is defined as the mean GRP/GEO, whereas s3 is defined as the mean GSP/GEO, but the Reviewer should pay attention that **all these measures including s2 and s3 are computed and defined only for the successful GRP paths!** Conversely, all the measures of congruence and navigability defined in our study are not constrained to be computed for only node pairs with a successful GRP. The measures proposed in our study are general and exact because they are defined taking in consideration all (nonadjacent) node pairs, regardless that a GRP is successful connecting them. Is this making a big difference? Yes, of course, because as Krioukov et al. admit in the same study, page 82:

<< These GF processes can be very inefficient. They can often get stuck at local minima, or even if they succeed reaching the destination, they can travel along paths much longer than the optimal shortest paths available in the network.>>

Indeed, in the new revised Fig1b,c (below we report both the new figure and the new figure legend) of our manuscript we offer two examples in which the GRP is successful and unsuccessful respectively, however in both cases the GRP (yellow line) is very different from the GSP (green line).

Then, how to interpret the results provided by Krioukov et al. in Fig.9 of their article, in which s2 and s3 offers the same results and therefore it seems that GRP is overlapping with GSP? They are obtained considering successful GRPs only and in hyperbolic networks generated in a idealized scenario where the temperature of the model $T=0$ (we stress that zero temperature is an idealized scenario that is far from happening in real hyperbolic networks). **In conclusion, s3 was used by Krioukov et al. to compare with s2 and assess the extent to which successful GRP have a path close to GSP, because they are comparably far from the GEO. However, s2 and s3 are not a general measure designed to assess congruence (indeed, Krioukov et al. never claimed this in their study) of a network topology with the geodesic because it neglects all the portion of the topology that is not explored by successful GRPs.**

Nevertheless, we believe that anytime a Reviewer rises a concern, regardless of the fact that is valid or invalid, it points out a possible weakness of the manuscript. In this case we honestly believe the responsibility of the Reviewer's confusion is clearly ours, because we were not able to write the previous version of the manuscript in a way that could avoid any misunderstanding to a reader. To address this issue, we implemented the following **two modifications in the revised manuscripts:**

(Modification 1) we add a new subsection at the beginning of the Results section which is dedicated to introduce all the types and differences between the paths considered in

the study. In association with this, we improve the Fig. 1 that now offers more examples to visually display the differences between the different types of paths (such as GRP, GSP and all the others) in accordance to what we comment in this new subsection.

The new sub-section is entitled: << **Types of paths and associated lengths in the topological and geometrical domains** >>, it is reported starting from Page 4 of the revised manuscript, and we invite the Reviewer to refer directly to the text in the revised manuscript for her/his check.

The new Fig. 1 is reported below. The Reviewer can notice the remarkable difference between GRP (yellow line) and GSP (green line) in panel b ad c.

Figure 1. Visualization of the different types of paths in the geometrical (hyperbolic) space.

A nPSO network has been generated with parameters $N = 100$, $d^- = 8$, $T = 0.1$, $\gamma = 2.5$ and $C = 5$. (a) Representation of the nPSO network in the hyperbolic disk: the links are in grey colour and follow the hyperbolic geodesics, the nodes are coloured by community membership and their size is proportional to the logarithm of the degree. (b) The panel highlights the hyperbolic geodesic (GEO, red), the geometrical shortest path (GSP, green), the greedy routing path (GRP, yellow with arrows of directionality) and the links involved in all the possible topological shortest paths (TSP, black dashed) between two specific nonadjacent nodes in the network (i and j), characterized by a successful GRP. (c) Analogous to panel (b) but for two nonadjacent nodes characterized by an unsuccessful GRP, which is highlighted in the zoomed subpanel. (d) The histogram indicates, at different intervals, the proportion of nonadjacent node pairs within a certain range of $GEO/(pTSP)^-$ values in the nPSO network. (e) The panel highlights GEO, GSP and TSP between two nonadjacent nodes characterized by a ratio $GEO/(pTSP)^- = 0.61$ corresponding to the mode value. (f) Analogous to panel (e) but for two nonadjacent nodes characterized by a ratio $GEO/(pTSP)^- = 0.67$ corresponding to the mean value. (g) Analogous to panel (e) but for two nonadjacent nodes characterized by a ratio $GEO/(pTSP)^- = 0.96$ corresponding to a value in the top 5% of the histogram.

(Modification 2) Right after the new subsection on the types of paths, we add another new subsection in the Results section that is dedicated to discuss the greedy routing navigability problem and its relation to congruence: << **From ‘myopic’ transfer and greedy navigability to the geometrical congruence** >>. This section comes with an extended Suppl. Inf. section that is about: << **Suppl. Inf. section 1: Previous measures for evaluation of greedy routing navigability and their limitations** >>. We invite the Reviewer to refer directly to the text in the revised manuscript for her/his check.

These new subsections of the revised manuscript aim to address the doubts that the Reviewer and any reader might have about paths and navigability, and clarify the previous measures adopted by Krioukov et al. and the associated results, as well as the way Krioukov and colleagues commented them. We hope that with these new sections any type of doubt or misinterpretation is dispelled.

The geometrical congruence (GS) introduced here is for all practical purposes GSP. This removes one of the main novel contributions in the manuscript.

REPLY: **this comment is invalid** because neither the definition of GSP or GRP (that we report in the previous section) matches with the one of GC (the Reviewer wrote GS clearly for a typo) or with the definition of $\overline{pTSP}(i, j)$. Indeed, the ‘main ingredient’ in the definition of GC is to account for all network topology by computing the mean projections of all topological shortest paths between pairs of nodes in a network $\overline{pTSP}(i, j)$, whereas the reference in respect to which this is compared can be any reference distance RD.

We define the geometrical congruence (GC) at Page 11 of our manuscript as: << Given a network with n nodes and e edges, we define the geometrical congruence (GC) as:

$$GC(\overline{pTSP}, RD) = \left(\frac{2}{n \cdot (n - 1) - 2 \cdot e} \right) \cdot \sum_{i < j} \frac{RD(i, j)}{\overline{pTSP}(i, j)} ; \text{ with } (i, j) \in \tilde{E}$$

where \tilde{E} is the set of pairs (i, j) of nonadjacent nodes. >>

Note that $RD(i, j)$ can be any node pairwise reference distance (not necessarily restricted to the geodesic) that is associated to the geometry. For instance, in this study we consider $RD(i, j)$ equal to the geodesic (GEO) in one case and to the geometrical shortest path (GSP) in the second case.

The statement of the Reviewer that << The geometrical congruence (GS) introduced here is for all practical purposes GSP. >> is as a matter of fact not understandable, because the definition of GC and GSP (or GRP) are not comparable. The 'main ingredient' to compute GC that makes possible to account for all network topology is $\overline{pTSP}(i, j)$, and not GSP (or GRP). Most importantly, and differently from s2 and s3, our evaluation is performed for all (nonadjacent) node pairs, whereas s2 and s3 proposed by Krioukov et al. only for the node pairs for which a successful GRP exists.

On this evidence, we reject the argument of the Reviewer that we do not introduce novelty. The novelty is in four points.

- (i) the general definition of geometrical congruence (GC) based on the $\overline{pTSP}(i, j)$.
- (ii) the efficient algorithm to compute our measures on large size networks (that the Reviewer declares to find of interest in the next comment).
- (iii) our measures account for all (nonadjacent) node pairs and therefore explore all the network topology, differently from the measures of Krioukov et al. that account only for node pairs with successful GRP.
- (iv) the results that we offer in evaluation of congruence and greedy navigability of hyperbolic networks, which are a correction and advancement of the current belief.

The reasons behind this novelty are explained in a new subsection (Page 7) in the Results section which is dedicated to discuss the greedy routing navigability problem and its relation to congruence: << **From 'myopic' transfer and greedy navigability to the geometrical congruence**>> This section comes with an extended Suppl. Inf. section that is about: << **Suppl. Inf. section 1: Previous measures for evaluation of greedy routing navigability and their limitations** >>. We invite the Reviewer to refer directly to the text in the revised manuscript for her/his check.

The algorithm for computing shortest paths might be of interest, as is analysis of the brain connectome. However, the manuscript need to be shorten by a factor of two at least to focus the exposition to novel results.

REPLY: we thank the Reviewer for recognizing the novelty and interest in the algorithm we proposed for efficiently finding all the possible shortest paths between the node pairs of a network and its application on brain connectomes. However, we stress that the

application on brain connectomes is a case study that shows how to apply our measures of congruence to networks with latent geometry. The other case study is on how to apply our measure on networks with patent geometry and it offers relevant results that we would prefer to share with the scientific community.

2) In the analysis of brain connectome data, the authors show that the latent space is not a three-dimensional Euclidean space. An obvious question that is left unanswered is whether it is hyperbolic.

REPLY: We thank the Reviewer to offer us the opportunity to clarify this point. As the Reviewer commented above, our current manuscript is already too long, therefore there is not space to introduce other topics that are not related directly with the hypothesis and questions on which the current study is focused. The measure of congruence and navigability discussed in our study are model-free and therefore do not make any assumption on the type of underlying geometry of the complex networks. For this reason, they cannot be directly employed to determine the type of geometry of the complex networks. This might require to build a null geometrical model, which is not the topic of this study. There is not anything that is misleading with the research proposed in the current version of the manuscript, which is not designed to reply to the question of what kind of geometry is behind a network, and to the best of our knowledge there is not yet any general test or measure that is able to convincingly reply to this question. This would require another study. In the current version of the manuscript we properly referred to the studies that are focused on understanding the geometrical nature of brain connectomes and this should be enough to fairly discuss this point, because we never claimed in a misleading way that in our study we were applying congruence and navigability to infer the latent geometry. However, we understand the request of the Reviewer to clarify this point even more explicitly, therefore we modified the manuscript adding the following text in bold.

Page 25, Results section on: Geometrical congruence in networks with latent geometry: the case of brain connectomes.

<< Finally, all results in Fig. 6 suggest that NOS is a better link weight dissimilarity in respect to 3D. Hence, the reply to the third question is yes: navigability and congruence offer evidence at support of the thesis that macroscale structural MRI brain connectomes might have a latent developmental neurobiological geometry (waiting for more evidences about whether it is hyperbolic²⁸⁻³⁰) which accounts for more information content than the 3D visible Euclidean geometry. **However, to avoid misunderstandings, we clarify that the measures of congruence and navigability discussed in this study are model-free and therefore do not make any assumption on the type of underlying geometry of the complex networks. For this reason, they cannot be directly employed to determine the type of geometry behind complex networks. This might require to**

build a null geometrical model, to use as a reference, and with respect to which variations of congruence and navigability should be compared. This is not the topic of the present study, and we hope that future studies might investigate how to exploit congruence and navigability to infer the nature (elliptic, Euclidean, hyperbolic) of the latent geometry behind many real networks. >>

More minor comments follow:

The abstract states that “hyperbolic networks do not demonstrate high congruence” but does not specify congruence with what. If the authors find that hyperbolic networks are not congruent with themselves, then this indicates a problem with their analyses.

REPLY: we thank the Reviewer. We modified the abstract by adding the specification requested by the Reviewer.

<< hyperbolic networks do not show in general **high GC and efficient greedy navigability (GN) with respect to the geodesics.** >>

In particular, the networks are generated here in a specific way based on the authors’ prior published work.

REPLY: The Reviewer did not notice that in our study we consider not only hyperbolic nPSO networks (which displays tailored number of communities) but also hyperbolic classical PSO networks (whose number of communities is not predetermined). As the Reviewer commented, the nPSO networks are generated according to an algorithm based on our previous work, however the PSO networks are generated according to an algorithm based on previous work on Krioukov et al. The results are consistent using both types of networks, therefore there is not any type of specificity associated to the type of hyperbolic model used to generate the networks. In addition, regardless of the model used, the most important requirement is to generate hyperbolic networks and as a matter of fact both nPSO (C=4) and PSO (C=0) networks are hyperbolic. This is the part of the Results section of the previous manuscript, and it is retained in the new revised manuscript.

Page 12: << For each panel of the figure the solid line indicates that community organization is not imposed (C=0, this is equivalent to the classical uniform **PSO model**¹²) and the dashed line indicates that networks with 4 communities are considered (C=4). In general, these two different community organizations seem not relevant for our investigation and therefore they are not further discussed. >>

The abstract uses the term “latent geometry” without ever defining it. How this is different from the standard term “latent geometry” is not described sufficiently well. There is a

mention of this on line 143, arguing that this the contrary of latent in Latin. A mathematical definition of how patent differs from latent space would be helpful.

REPLY: We thanks the Reviewer for the recommendation. Introduction, page 6, we revised the current manuscript adding the following text in bold: << Then, we proceed to measure geometrical congruence and greedy navigability efficiency in networks with patent (the contrary of latent in Latin) geometrical space, in the sense that the coordinates of the network nodes in this space are known; **as opposed to a latent geometrical space in which the coordinates of the network nodes are unknown.** >>

The introductory opening to the manuscript (line 38-47) is not relevant to the science described in the manuscript. It would be better to use a different opening that does not use the words “hatred” and “treachery”.

REPLY: We thanks the Reviewer for this comment. We modified the Introduction, Page 3, adding the following text in bold: << “There is enough treachery, hatred violence absurdity in the average human being to supply any given army on any given day [...]”. **Although feelings such as treachery and hatred do not comply with the scope of this study**, in the incipit of the poem ‘The Genius of the crowd’, Charles Bukowski offers an impressive lecture on how to leverage the average for analysing collective behaviours in a complex disordered system such as our society, [...]

Similarly, the term “memoization” introduced as part of the proposed algorithm has a very specific meaning in computer science. It would be better not to use it here. Memorization would be easier for readers and more appropriate

REPLY: We understand the point of the Reviewer. We explained the term “memoization” in the Method section, but we did not clarify this in the Results section and we agree with the Reviewer that this might induce confusion for non-technical readers. On the other side, the term memorization is not technically precise but it is not wrong. We agree that would be easier for readers to also consider the term memorization. For this reason, we added to the manuscript, in the first point in which we introduce the term memoization, the following text in bold:

Page 15, Results section on: An algorithm to compute geometrical congruence in networks.

<< The term memoization has a very specific meaning in computer science that we discuss in depth in the respective method section associated to this algorithm. For non-technical readers, this key innovation can be viewed as a memorization strategy ensuring that a function does not run for the same inputs more than once, by keeping a record (memory) of the results for the given inputs. >>

Line 514, instead of “two information” it might be better to state “two pieces of information” or “two properties”.

REPLY: we thank the Reviewer, we modified the text according to his suggestion: “**two pieces of information**”.

Line 537: it is not appropriate to refer to the authors own work as "the first" given that these publications (refs. 22 and 23) are from 2017 and 2019 whereas Krioukov and collaborators have been publishing on this topic since at least 2008.

REPLY: When we wrote this text: << Muscoloni et al.^{22,23} were the first to design a measure of network greedy navigability efficiency for complex networks in general, and Seguin et al.³² were the first to discuss and motivate its application on brain connectomes. >> We used the word first to indicate that our group was the first to propose a measure of efficiency for complex networks in general (whereas Krioukov et al. proposed a measure in hyperbolic networks only as we clarified in the new subsection on navigability) and Seguin et al.³² then proposed the same measure of efficiency of Muscoloni et al. applying it to brain connectomes and they did not refer to us. Because of this, PNAS decided to publish our letter to recognize the merit that we were the first to devise a measure of greedy routing efficiency in complex networks in general (that accounts for all GRPs – successful and unsuccessful - and is not restricted to hyperbolic networks):

Navigability evaluation of complex networks by greedy routing efficiency

A Muscoloni, CV Cannistraci

Proceedings of the National Academy of Sciences 116 (5),2020, 1468-1469

However, we understand the concern of the Reviewer and we addressed his requirement modifying the sentence in this way:

<< The second is an important open problem in network neuroscience³⁴. **Several measures of greedy navigability were proposed by Krioukov et al. for hyperbolic networks, such as hyperbolic stretches¹¹, but they were supposed to work primarily in hyperbolic networks. Unfortunately, we offered evidence that even in hyperbolic networks these measures show some inconsistencies, because of the limitations reported in the Results section of this study that is dedicated to navigability. Therefore, to address such issues, Muscoloni et al.^{22,23} designed a measure of network greedy navigability efficiency for complex networks in general (not only hyperbolic), and successively Seguin et al.³⁴ discussed and motivated its application on brain connectomes.** Here we wish to make a step forward to investigate whether greedy navigability efficiency, and consequently also geometrical congruence, can be used as markers to differentiate different phenotypical states in brain connectomes. >>

Reviewer #2 (Remarks to the Author):

I read the revised version of the paper that in my opinion has greatly improved. The authors answered with great detail to my questions. I am satisfied by their work. I ask publication of the paper in the present form

REPLY: we are grateful to the Reviewer for the effort she/he put to help us to improve the manuscript.

Reviewers' Comments:

Reviewer #1:

Remarks to the Author:

The manuscript is not valid for the same two reasons as were described in the past review:

In my last review I wrote:

1) "One of the main results in the manuscript aims to disprove the results by Krioukov and collaborators that topologically shortest paths (TSP) are similar to geodesics in the underlying hyperbolic space. The authors show that TSP are not similar to geodesics but are similar to geometric shortest paths (GSP). However, it could be argued that GSP is the metric that Krioukov used. The algorithm there was to find a neighboring node with the shortest hyperbolic distance and iterate until the destination node is reached. This is the algorithm behind the GSP used in this paper. Therefore, the main result in this manuscript disputes not the finding by Krioukov but rather the misinterpretation of it."

The authors write that my comment is not valid because the procedure that I "describe corresponds to GRP and not GSP". However, further below in their response the authors acknowledge that Krioukov et al used *both* GRP and GSP measures.

2) The second reason stated in the past review was that the geometrical congruence (GC) introduced here is for all practical purposes GSP.

Although the authors labeled this comment as invalid, they acknowledge that in their equation for GC, the pairwise reference distance "RD(i,j) equal to the geodesic (GEO) in one case and to the geometrical shortest path (GSP) in the second case."

Reviewer #3:

Remarks to the Author:

1) On the debate between Reviewer #1 and the authors

According to the reports, Reviewer #1 had two comments that the authors claimed to be invalid. Although a detailed argumentation was provided in the reply by the authors, Reviewer #1 was not convinced. In general, I am on the side of the authors in this debate, and in my opinion, these comments were indeed not well founded, and accordingly, should not detain the acceptance of the paper.

In the first comment, Reviewer #1 argued that the GSP was already used in an earlier paper by Krioukov et al., and also stated that parts of the findings in the present manuscript come from a misinterpretation of the results in that paper. Related to this, I agree with the authors in that the most essential difference between the earlier results and the present analysis is that here the paths between all possible pairs are considered, whereas in the paper by Krioukov et al., only the successful greedy paths were taken into account. Besides that, the question of whether a new kind of path length is introduced here or not is of less relevance in my opinion. Furthermore, during the revision, the authors did put a great effort into explaining how their new findings correct and refine the previous results in the literature. Finally, in my interpretation, the quoted part from the paper by Krioukov et al. may also refer to pTSP instead of GSP, hence it is not clear whether GSP was really used in this earlier work.

In the second comment, Reviewer #1 suggested that "GC is for all practical purposes GSP". Since GC is a measure for comparing two sets of distances, whereas GSP is a "simple" distance measure, in my opinion, this statement does not make any sense, and I cannot really work out what the Reviewer

really meant by that.

According to the above, as stated in the beginning, I agree with the authors that these comments are invalid.

2) Opinion about the manuscript

In general, I find the paper very interesting and the results worth publishing. Meanwhile, I also have a remark that the authors might take into consideration before the final release:

In Fig.1. the links do not follow the geodesics. The curved lines shown in this figure seem to be (parts of) circles (that would arrive at a right angle to the disk perimeter), which correspond to the correct geodesics on the Poincaré disk representation of the hyperbolic space. However, the nPSO model is defined on the native disk representation of the hyperbolic space where the geodesic lines do not follow circles. For illustration please see the figure attached to this report, displaying an nPSO network with 100 nodes where the links are drawn according to the geodesic lines on the native disk.

I would suggest either replacing the circular curves with the true geodesics in Fig.1., or if this makes the images messy, then at least indicate in the caption that the links are curved only for illustrative purposes and do not follow the geodesics.

REPLY TO REVIEWER COMMENTS

Reviewer #1 (Remarks to the Author):

The manuscript is not valid for the same two reasons as were described in the past review:

In my last review I wrote:

1) “One of the main results in the manuscript aims to disprove the results by Krioukov and collaborators that topologically shortest paths (TSP) are similar to geodesics in the underlying hyperbolic space. The authors show that TSP are not similar to geodesics but are similar to geometric shortest paths (GSP). However, it could be argued that GSP is the metric that Krioukov used. The algorithm there was to find a neighboring node with the shortest hyperbolic distance and iterate until the destination node is reached. This is the algorithm behind the GSP used in this paper. Therefore, the main result in this manuscript disputes not the finding by Krioukov but rather the misinterpretation of it.”

The authors write that my comment is not valid because the procedure that I “describe corresponds to GRP and not GSP”. However, further below in their response the authors acknowledge that Krioukov et al used *both* GRP and GSP measures.

REPLY: we thank the Reviewer 1 for the effort she/he put to help us in improving the manuscript. Many results we provide in this article are merit of the debate we are having together. However, we want to stress again that the use of GSP in Krioukov et al. is totally different from the use of GSP in our study, and that Krioukov et al. never took in consideration to propose a measure of congruence that considers the mean projection of all the TSPs between nonadjacent nodes (i,j) in a network.

2) The second reason stated in the past review was that the geometrical congruence (GC) introduced here is for all practical purposes GSP.

Although the authors labeled this comment as invalid, they acknowledge that in their equation for GC, the pairwise reference distance “RD(i,j) equal to the geodesic (GEO) in one case and to the geometrical shortest path (GSP) in the second case.”

REPLY: GC is a measure that is based on the ratio between two sets of distances, whereas GSP is a pure distance measure. Again, we cannot understand what the matter of the Reviewer’s concern is.

Reviewer #3 (Remarks to the Author):

1) On the debate between Reviewer #1 and the authors

According to the reports, Reviewer #1 had two comments that the authors claimed to be invalid. Although a detailed argumentation was provided in the reply by the authors, Reviewer #1 was not convinced. In general, I am on the side of the authors in this debate, and in my opinion, these comments were indeed not well founded, and accordingly, should not detain the acceptance of the paper.

REPLY: We thank the Reviewer 3 for recognizing the contribution of our scientific work.

In the first comment, Reviewer #1 argued that the GSP was already used in an earlier paper by Krioukov et al., and also stated that parts of the findings in the present manuscript come from a misinterpretation of the results in that paper. Related to this, I agree with the authors in that the most essential difference between the earlier results and the present analysis is that here the paths between all possible pairs are considered, whereas in the paper by Krioukov et al., only the successful greedy paths were taken into account. Besides that, the question of whether a new kind of path length is introduced here or not is of less relevance in my opinion. Furthermore, during the revision, the authors did put a great effort into explaining how their new findings correct and refine the previous results in the literature. Finally, in my interpretation, the quoted part from the paper by Krioukov et al. may also refer to pTSP instead of GSP, hence it is not clear whether GSP was really used in this earlier work.

REPLY: We thank the Reviewer 3 for studying so carefully the matter of the debate and for recognizing our scientific contribution.

In the second comment, Reviewer #1 suggested that "GC is for all practical purposes GSP". Since GC is a measure for comparing two sets of distances, whereas GSP is a "simple" distance measure, in my opinion, this statement does not make any sense, and I cannot really work out what the Reviewer really meant by that.

REPLY: We, as the Reviewer 3, could not really work out what the Reviewer 1 really meant with this concern, but it seems to us that it is invalid.

According to the above, as stated in the beginning, I agree with the authors that these comments are invalid.

REPLY: We thank the Reviewer 3 for independently recognizing that the comments of Reviewer 1 are invalid.

2) Opinion about the manuscript

In general, I find the paper very interesting and the results worth publishing.

REPLY: We thank the Reviewer 3 for recognizing the value and interest of our research.

Meanwhile, I also have a remark that the authors might take into consideration before the final release:

In Fig.1. the links do not follow the geodesics. The curved lines shown in this figure seem to be (parts of) circles (that would arrive at a right angle to the disk perimeter), which correspond to the correct geodesics on the Poincaré disk representation of the hyperbolic space. However, the nPSO model is defined on the native disk representation of the hyperbolic space where the geodesic lines do not follow circles. For illustration please see the figure attached to this report, displaying an nPSO network with 100 nodes where the links are drawn according to the geodesic lines on the native disk.

I would suggest either replacing the circular curves with the true geodesics in Fig.1., or if this makes the images messy, then at least indicate in the caption that the links are curved only for illustrative purposes and do not follow the geodesics.

REPLY: We thank the Reviewer 3 for providing such a useful comment. We opted to redesign the Fig. 1 using the native disk representation, replacing the circular curves with the true geodesics. In order to overcome the problem raised by the Reviewer 3 (“if this makes the images messy,...”), we used special side-panels where we provided an exploded-view representation of the paths. This solution allowed to visualize better the trajectories of the paths.